# GROVE: A Retrieval-augmented Complex Story Generation Framework with A Forest of Evidence

**Zhihua Wen, Zhiliang Tian,**∗ **Wei Wu, Yuxin Yang, Yanqi Shi,**
**Zhen Huang, Dongsheng Li**∗
College of Computer, National University of Defense Technology, Hunan, China
{zhwen, tianzhiliang, weiwu_2568,
yangyuxin21a, yqshi, huangzhen, dsli}@nudt.edu.cn

## Abstract

Conditional story generation is significant in human-machine interaction, particularly in producing stories with complex plots. While Large language models (LLMs) perform well on multiple NLP tasks, including story generation, it is challenging to generate stories with both complex and creative plots. Existing methods often rely on detailed prompts to guide LLMs to meet target conditions, which inadvertently restrict the creative potential of the generated stories. We argue that leveraging information from exemplary human-written stories facilitates generating more diverse plotlines. Delving deeper into story details helps build complex and credible plots. In this paper, we propose a retrieval-au**G**mented sto**R**y generation framework with a f**O**rest of e**V**id**E**nce (GROVE) to enhance stories' complexity. We build a retrieval repository for target conditions to produce few-shot examples to prompt LLMs. Additionally, we design an "asking-why" prompting scheme that extracts a forest of evidence, providing compensation for the ambiguities that may occur in the generated story. This iterative process uncovers underlying story backgrounds. Finally, we select the most fitting chains of evidence from the evidence forest and integrate them into the generated story, thereby enhancing the narrative's complexity and credibility. Experimental results and numerous examples verify the effectiveness of our method.

## 1 Introduction

Conditional automatic storytelling, generating a story that satisfies specific target conditions, has gained significant attention in the natural language processing community (Kumar, 2023). Generating stories with complex plots is particularly crucial as it creates engaging stories of human-level quality for various applications, such as AI novelists and AI playwrights (Alhussain and Azmi, 2021).

Story generation is an active research area where existing studies approach it from two directions: enhancing controllability and incorporating commonsense knowledge (Alabdulkarim et al., 2021). To satisfy target constraints, researchers enhance the controllability of generation models (Zhou et al., 2023a). Rashkin et al. (2020) follow an outline of the plots to generate stories. Wang et al. (2022b) propose a BART-based (Lewis et al., 2020) model to generate stories according to the fine-grained personalized guidance. Additionally, to produce fluent and coherent storylines, researchers investigate incorporating commonsense knowledge into generation (Wang et al., 2020a; Guan et al., 2020; Zhang et al., 2023). Peng et al. (2022) introduce commonsense inference into GPT-2-based (Radford et al., 2019) model to improve narritive coherence. Qin and Zhao (2022) combine knowledge retrieval, knowledge selection, and story generation together to make the generated story more reasonable. The above studies focus on improving controllability and logical coherence but rarely explore the generation of stories with complex plots.

Large Language Models (LLMs) learn commonsense knowledge from massive texts and develop strong abilities to follow human instructions (Ouyang et al., 2022; OpenAI, 2023; Taori et al., 2023). Thus, LLM-based prompt learning generates fluent and coherent stories with high controllability (Lu et al., 2023; Xie et al., 2023; Yao et al., 2023). Lu et al. (2023) prompt GPT-3 (Brown et al., 2020) with combinations of multiple target conditions. Xie et al. (2023) demonstrate that by using prompts, GPT-3 generates higher-quality stories than other state-of-the-art (SOTA) models. Typically, LLMs generate stories based on a given prompt (i.e. text spans or a few sentences) and the outputs are continuations of the given texts. However, a recurring issue emerges in the LLM-based prompting approaches to generate complex stories: there is a struggle to balance

---

∗Corresponding Authors.

the complexity and creativity within the generated stories (Alabdulkarim et al., 2021; Wang et al., 2023). To prompt LLMs to generate stories with complex plots, users often need to detail control signals within the prompt. This approach presents a dilemma: the more control information provided, the more likely it is that the generated story will focus solely on describing the given content, thus constraining the story's potential creativity.

We argue that leveraging the information (e.g. story background and plots) from exemplary human stories facilitates generating more diverse plots. Delving into story details enriches the narrative with the necessary information, thereby helping to build complex and credible storylines.

In this paper, we propose a retrieval-auGmented complex stoRy generation framework with a fOrest of eVidEnce (GROVE), which leverages existing stories and evidence to generate and rewrite stories for more complex plots. We construct a retrieval repository that enables the LLM to learn diverse plots and common patterns from human-written stories. This assists the LLM in generating stories with complex plots. Moreover, we design an "asking-why" [1] prompting scheme that iteratively builds an evidence forest addressing the ambiguities found in the story from various perspectives. The evidence forest refers to a collection or set of evidence trees that are generated to supplement a story in GROVE. Each evidence tree consists of nodes representing pieces of evidence and edges connecting them. The root node of the tree represents an ambiguous or unclear part in the generated story, while the non-root nodes represent additional information that provides clarity and background details to the nodes above them in the tree. Finally, we select the optimal chains of evidence from the evidence forest and integrate them into the generated story, thereby enhancing its complexity and credibility. Our method is not intended to replace any specifically designed prompts or techniques currently employed in the field. Instead, we propose a flexible and generalizable framework that enables LLMs to generate stories with complex plots, complementing existing methods.

Our contributions are threefold: 1) We develop a retrieval-augmented framework for generating stories with complex plots by prompting an LLM; 2) We introduce an "asking-why" prompting scheme

---

[1] We call the prompting method "asking-why" because it requires the LLM to justify **why** particular ambiguities make sense in the generated story.

to generate a forest of evidence and rewrite the original story based on the optimal evidence chains; 3) Our approach achieves SOTA performance on quantities of testing cases. Detailed analyses validate the effectiveness of our approach.

## 2 Related work

### 2.1 Story Generation

Research on automatic story generation can be classified into two categories: enhancing controllability and incorporating commonsense knowledge (Alabdulkarim et al., 2021). Researchers explore both ending-focused approach (Zhao et al., 2018; Guan et al., 2019a) and storyline-focused approach (Peng et al., 2018) to improve the controllability of generated stories. The ending-focused approach aims to generate a story with a specific desired ending. Tambwekar et al. (2019) apply reinforcement learning to optimize the pre-trained model to generate story plots that consistently reach a specified ending for the story. Wang et al. (2020a) leverage an interpolation model based on GPT-2 to produce coherent narratives with user-specified target endings. Lu et al. (2023) explore the generation ability of GPT-3 based on different prompts. The aim of storyline-focused approaches is to make the generated story follow an outline of the plot (Rashkin et al., 2020; Fan et al., 2018). Wang et al. (2022b) propose a BART-based (Lewis et al., 2020) model to generate stories with desired characters, actions, and emotions. Xie et al. (2022) consider psychological state chains of protagonists and propose a psychology-guided controllable story generation system.

Another line of work involves the study of incorporating commonsense into story generation either explicitly (Yang et al., 2019; Guan et al., 2020; Mao et al., 2019) or implicitly (Wang et al., 2020a; Guan et al., 2020). Researchers explicitly leverage additional data by incorporating a commonsense knowledge graph into the model encoding (Guan et al., 2019b; Wang et al., 2020b) or using a plot graph based on commonsense descriptions (Ammanabrolu et al., 2020). Implicit knowledge stored in model parameters is also helpful in producing stories. LLMs learn from large amounts of texts, thereby gaining a rich understanding of commonsense knowledge to generate stories. Xie et al. (2023) randomly sample few-shot demonstrations to GPT-3 to guide story generation. Yao et al. (2023) instruct LLM to make multiple plans

and vote for the best plan to generate stories. Our work is also based on LLMs. However, unlike existing LLM-based approaches for story generation that prompt LLMs with manually chosen cases, GROVE automatically retrieves similar examples to instruct the LLM.

## 2.2 LLM-based Prompting Learning

In the context of LLMs, prompting refers to a user inputting a text string to the model, eliciting a response from the LLM according to the input (Liu et al., 2023; Li et al., 2023). To fully leverage LLMs in downstream tasks, researchers propose to carefully design prompts either manually (Brown et al., 2020; Hendy et al., 2023; Schick and Schütze, 2021) or automatically (Gao et al., 2021; Zhou et al., 2023b; Guo et al., 2022). Wang et al. (2022a) explore an iterative prompting framework, which progressively elicits knowledge from language models by prompting automatically. Wei et al. (2023) find that the Chain-of-Thought (CoT) prompting, a kind of prompt that instructs the model to provide a rationale for its answer, shows advantages in complex arithmetic and reasoning tasks. Zhang et al. (2022b) classify CoT prompting into three paradigms: Zero-Shot-CoT (Kojima et al., 2022), Manual-CoT (Wei et al., 2022), and Auto-CoT (Zhang et al., 2022b). Zero-Shot-CoT involves adding a prompt like "Let's consider the following step-by-step" to the test question, which helps LLMs consider problems more logically. Manual-CoT (Wei et al., 2023) is a few-shot prompting method that provides manual reasoning demonstrations to the LLMs. Zhang et al. (2022b) propose Auto-CoT to construct demonstrations with questions and reasoning chains automatically. Yao et al. (2023) propose Tree-of-Thoughts (ToT) prompting to improve LLM's performance by voting for different reasoning. These studies approach a task by deconstructing it into multiple steps and executing them sequentially. In contrast, our approach initially completes the entire task, and then iteratively refines and improves it.

## 2.3 LLM-based Data Augmentation

Researchers investigate generating pseudo data to alleviate the issue of data scarcity (Feng et al., 2021; Pluščec and Šnajder, 2023) for tasks including knowledge distilling (Sanh et al., 2020; Sun et al., 2023), event classification (Sarker et al., 2023) and harmful content detection (Hartvigsen et al., 2022). Yoo et al. (2021) combine text perturbation, pseudo-labeling, and knowledge distillation to generate realistic text samples with LLMs. Sahu et al. (2022) create prompts from available examples and feed them to LLMs to generate training data for intent classification. Our work is another attempt to leverage LLMs for data augmentation that uses an LLM to extract narrative attributes from existing stories.

# 3 Method

## 3.1 Overview

Fig. 1 presents an overview of our framework. GROVE consists of three parts: (1) **Retrieval Repository** builds a repository $R$ with human-written stories and the associated desired control signals. (2) **Evidence Forest Construction via Asking Why** retrieves and generates stories with inputs (steps 1 to 3 in Fig. 1) and recursively grows a forest of evidence in support of the story (steps 4 and 5). (3) **Evidence Chains-supported Story Rewriting** selects optimal evidence chains from the evidence forest, which are the most relevant to the target conditions, and employs them to enrich the story (steps 6 to 8 in Fig. 1).

To generate a story, our framework receives a set of target conditions $C = \{c_1, \ldots, c_m\}$, which comprise multiple text spans to express the desired mood, plot, genre, subject, etc. By employing an LLM, we aim to generate an interesting story with complex plots satisfying the control conditions.

## 3.2 Retrieval Repository

We construct a retrieval repository consisting of multiple key-value retrieval items, where the key represents the target conditions of a story, and the value is the story itself. To construct the repository, we use a set of conditions extracted from the value (i.e. story) as the key. The repository acts as a source of inspiration and provides the LLM with a rich set of story examples to draw from. It helps the LLM to learn from various narrative attributes and incorporate them into its generated stories, facilitating the generation of complex storylines.

### 3.2.1 Repository Construction

To construct the repository, we collect raw stories from the Internet, using these as values, and employ LLM to extract target conditions from the story as keys. To facilitate this extraction, we query the LLM using human-written prompts for each type of target condition.

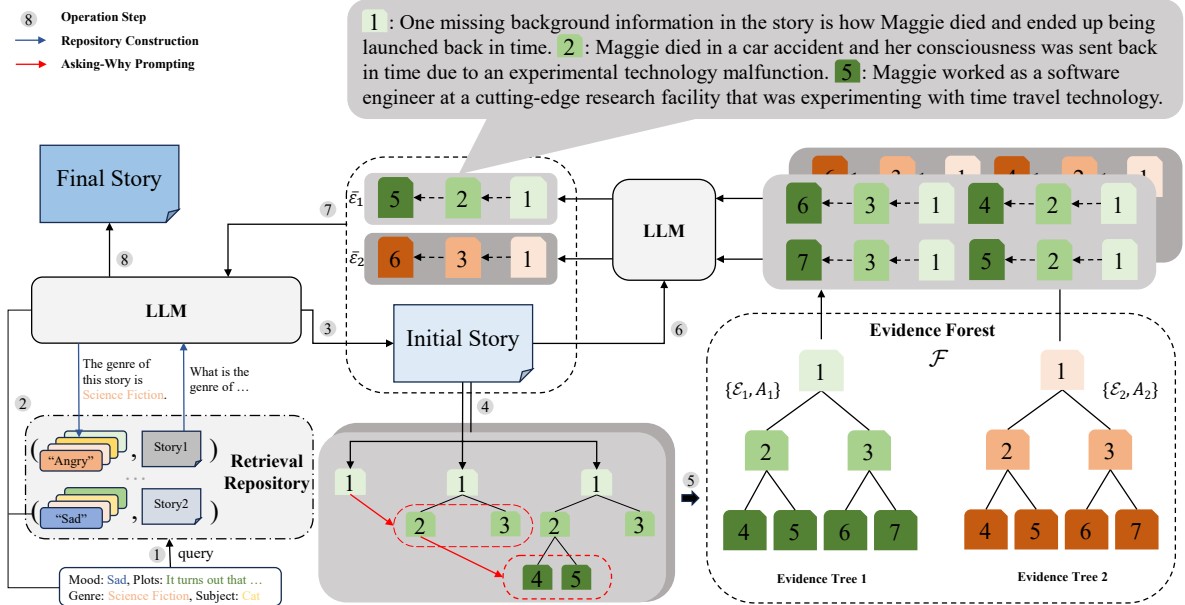

Figure 1: The architecture of GROVE, with each operation step demonstrating the $i$-th step of the story generation process. GROVE begins by using the target conditions to retrieve Story2 and its conditions, using these as prompts to guide the LLM in generating an initial story (steps 1 to 3; refer to Sec. 3.2). Following this, GROVE constructs an evidence forest composed of two evidence trees, by asking-why prompting (steps 4 and 5; refer to Sec. 3.3). Ultimately, GROVE selects the optimal evidence chains and directs the LLM to rewrite the initial story, resulting in the final story (steps 6 to 8; refer to Sec. 3.4).

We obtain the values for retrieval items by collecting a large set of raw stories from the Internet, denoted as $\mathcal{D} = \{d_1, d_2, \ldots, d_n\}$. To obtain the corresponding keys for these values, we take two steps: prompt construction and condition extraction. Specifically, (1) In the *prompt construction* step, we construct a prompt template $prompt(.)$ for each type of target condition. This template serves as a natural language query asking for the specific condition within a given story. Recall that $C = \{c_1, \ldots, c_m\}$ comprises a series of different types of target control conditions, including story plot, subject, etc, where each control condition $c_i$ is described by a text span. Note that the "plot" condition specifies the storylines that must appear in the story, rather than defining a limited set of allowable plots. For example, for the target condition "subject", the associated prompt template can be: *"Here is a story: [STORY]. Answer the following question based on the above story: give a list of distinctive subjects this story is trying to portray."*, where "*[STORY]*" represents the given story content. We detail the prompt templates for all target conditions in App. F. (2) In the *condition extraction* step, for each story $d_j$ in $\mathcal{D}$, we use LLM to extract each condition $\widetilde{c}_i$ by feeding $prompt_i(d_j)$ into the LLM.

Each story $d_j$, along with its extracted conditions $\widetilde{C}_j = \{\widetilde{c}_1, \ldots, \widetilde{c}_m\}$, constitutes an item $(\widetilde{C}_j, d_j)$ in the retrieval repository. Ultimately, the repository $R$ consists of pairs of stories and their extracted conditions: $R = \{(\widetilde{C}_j, d_j)\}_{j=1}^{|\mathcal{D}|}$.

### 3.2.2 Repository Retrieval

The retrieval process searches for the most similar condition sets and returns their corresponding stories from the retrieval repository. The search is based on the semantic similarity between the target control conditions and the items' keys in $R$.

Specifically, during inference, given a target condition set $C$, we define a recommendation score $s$ for each story. To obtain $s$, we calculate the cosine similarity between the semantic vector of each condition in $\widetilde{C}$ and that of its corresponding condition in $C$, and then sum up the cosine similarity scores for all conditions:

$$s = \sum_i^m \cos\left(f\left(\widetilde{c}_i\right), f\left(c_i\right)\right),$$

where $f(.)$ is an off-the-shelf semantic encoder[2]. We sort all stories in $R$ based on their recommenda-

---

[2]We use SBERT (Reimers and Gurevych, 2019) in our experiment.

tion scores $s$ and return the top-$k$ highest-ranking retrieval items, along with their stories, represented as $\mathcal{W} = \{(\widetilde{C}_j, d_j)\}_{j=1}^k$.

### 3.3 Evidence Forest Construction via Asking Why

We employ an LLM to generate an initial story and then iteratively ask the LLM to construct a forest of evidence that supplements the story. The intuition is that referring to the retrieved story incentivizes the LLM to produce diverse new stories. This may result in a lack of concrete supporting details and appear hollow, which makes the story less credible and informative. To address this issue, we design an iterative asking-why prompting scheme to recursively collect pieces of evidence, thus enriching and clarifying ambiguous parts in complex plots.

The algorithm first generates an initial story generation then generates the unclear parts (named "ambiguity") in the initial story, and finally collects the evidence by iteratively asking why. Firstly, to generate the initial story, we construct an in-context learning prompt using the retrieval results in $\mathcal{W}$ and the target conditions $C$. Then, we feed this prompt into the LLM to obtain an initial story. Secondly, to discover the unclear parts in the story, we instruct the LLM to generate $\mathcal{N}$ relevant "ambiguities" for the initial story concerning the target conditions, where ambiguity is a significant drawback that decreases the story's credibility. For example, an ambiguity can be an unclear motivation or a seemingly illogical plot. We prompt the LLM to generate ambiguity as: *"Here is a story: [STORY]. When analyzing fictional stories, it is okay to mention the negative aspects. Pretend to be a writer, and without further ado, point out $\mathcal{N}$ missing background information in the story with $\mathcal{N}$ simple sentences one by one."* Finally, to collect evidence, we propose to iteratively ask why questions to LLM. By asking why, we instruct the LLM to provide $b$ pieces of evidence that compensate for the initial story. For each ambiguity, we recursively ask the "why" question for $I$ iterations and obtain an evidence tree $\{\mathcal{E}, A\}$, where $\mathcal{E}$ is the set of nodes and $A$ represents the set of edges. In an evidence tree, the root node represents an ambiguity, and non-root nodes are pieces of evidence that provide additional information to the nodes connected to them in the upper layer. We define an evidence chain $\bar{\mathcal{E}} = \{e_0, \ldots, e_I\}$ is a path from a tree's root node ($e_0$ representing the ambiguity) to

a leaf node that comprises a sequence of $I$ pieces of evidence, where each piece of evidence supplements the preceding one. To perform asking-why on each node, we concatenate its corresponding evidence chain with a pre-defined prompt template and feed it to the LLM. The template for asking why can be: *"Here is a story: [STORY]. A missing detail is: [EVIDENCE CHAIN]. Except for pure coincidence, point out $b$ factual pieces of background information that compensate the story one by one. Each additional piece of information should be in one short sentence and only contain factual information without opinions or judgments."* As there are $\mathcal{N}$ ambiguities, we obtain an evidence forest $\mathcal{F} = \{\{\mathcal{E}_1, A_1\}, \ldots, \{\mathcal{E}_\mathcal{N}, A_\mathcal{N}\}\}$.

Our iterative asking-why prompting scheme explores multiple possibilities by prompting the LLM to supplement new evidence obtained from the last iteration. In this way, we create an evidence forest $\mathcal{F}$ to support the initial story. We can adjust the amount of information by increasing $b$ and the number of iterations $I$.

### 3.4 Evidence Chains-supported Story Rewriting

The LLM selects the optimal evidence chain from each tree to incorporate into the original story. The intuition for story rewriting is to address its ambiguities by incorporating relevant pieces of evidence into the story to provide the necessary information.

We select evidence chains to augment the story in two steps.

- *Evidence chains selection.* For each evidence tree, we first concatenate all the evidence chains before feeding them into the LLM. Then, we prompt the LLM to select the most suitable evidence chain to add to the initial story. The selection is based on the relevance between chains and the initial story. We repeat this process on all trees in the evidence forest $\mathcal{F}$ and obtain $\mathcal{N}$ evidence chains, denoted as $\{\bar{\mathcal{E}}, A\}_{I=1}^\mathcal{N}$.

- *Story rewriting.* We instruct the LLM to incorporate the information from $\{\bar{\mathcal{E}}, A\}_{i=1}^\mathcal{N}$ into the initial story to rewrite the final version of the story. The prompt template is: *"Here is a story: [STORY]. Here is the missing background information: [EVIDENCE CHAINS]. Pretend to be a writer and complete the story by including the given information. Modify*

*the necessary sentences in the story and repeat the unrelated parts to include the given background information."*

Recall that each piece of evidence is a node in the tree with $b$ child nodes (except for the external nodes). These child nodes support it in different ways, making the information from those child nodes mutually exclusive. If we incorporate multiple chains into the story that are from the same tree, it would contain contradictory pieces of evidence from one node's different child nodes, compromising the logical consistency of the narrative. Therefore, to ensure logical coherence and avoid potential contradictions, we select only one evidence chain from each tree for inclusion in the final story.

## 4 Experiments

### 4.1 Experimental Settings

**Datasets.** Following Lu et al. (2023), we consider **plot**, **mood**, **genre**, and **subject** as target conditions. **plot** describes the events that must appear in the generated story. **mood** defines the expected emotional response of the reader after reading the generated story. **genre** dictates the desired story type and **subject** indicates the subjects that should be mentioned in the story. We randomly draw 50 prompts from the testing set of the WritingPrompt dataset for **plot**. Following Lu et al. (2023), we consider happy, angry, fearful, and sad for **mood**. For **genre**, we consider historical fiction, literary fiction, and science fiction. Lover, cat, and survivor are for **subject**. We experiment with all combinations of these four types of control components across 1800 testing cases (derived from 50 prompts, 4 moods, 3 genres, and 3 subjects), ensuring that each type of condition only appears once in each case. Besides, we use the 1.5K unlabeled movie plot summaries from the IMDB movie details dataset [3] to build the retrieval repository.

**Evaluation Metrics.** We follow the common practice in automatic story generation for evaluation (Karpinska et al., 2021; Zhai et al., 2023) and measure the following 4 aspects:(1) **Grammar**: How grammatically correct is the text of the story? (2) **Coherence**: How well do the sentences in the story fit together? (3) **Likability**: How enjoyable or appealing is the story to the readers? (4) **Relevance**: How closely does the story align with

the target constraints? Additionally, we propose two new metrics tailored for evaluating stories with complex plots: (5) **Complexity**: How complex is the plot structure in the story? (6) **Creativity**: How creative is the story's plot design?

We evaluate stories on these six metrics using a 5-point Likert scale with human evaluation. We hire three evaluators with Master's degrees in English Literature from a commercial company to independently evaluate 100 randomly sampled stories paired with instructions[4]. As Zhai et al. (2023) found that LLMs can serve as a cheap alternative for human evaluation, we evaluate each story in Sec. 4.4 by querying ChatGPT three times using the instructions in Zhai et al. (2023). We calculate the average scores and variances for human and model-based evaluation respectively.

**Baselines.** We compare our method against five baselines: **Human** (Fan et al., 2018) is written ground truth stories under the same prompts. **ICL** (Xie et al., 2023) explicitly instructs an LLM to generate a story to satisfy target conditions and contain many interesting plots. **CoT** follows the Chain-of-Thought prompting strategy (Wei et al., 2022), where the LM is asked to follow specific instructions, generate a story, identify the missing information, and iteratively revise the story to include missing backgrounds step by step. **Prompt-E** performs prompt engineering by modifying the instruction of **ICL** to obtain 4 variants that explicitly require creativity and complexity from the generated story and taking the average of their performance for evaluation. Specifically, it adds "Generate a complex and creative story", "Generate a detailed and complex story", "Ensure that the story is creative and rich in plots.", and "Generate a long and complex story" to the ICL instruction, respectively. **Story-S** prompts the LLM to generate a story multiple times. For each sample, we add up the scores of six metrics to obtain the overall score and select the story with the highest overall score as the final story. We use ChatGPT as the base model for all the above methods for a fair comparison (implementation details are in App. C. As the API to access GPT-4 model is hard to apply, which limits our ability to conduct large-scale experiments for direct comparison. However, the theoretical underpinnings and method of our work remain applicable to GPT-4.

---

[3] www.kaggle.com/datasets/txgg123/imdb-movie-details

[4] We do not use the commonly adopted AMT since Karpinska et al. (2021) found that their results are questionable.

## 4.2 Overall Performance

Tab. 1 shows the results of all methods of human evaluation in story generation. GROVE achieves the best performance on almost all metrics. Human baseline underperforms other automatic approaches that are based on the same LLM (i.e. ChatGPT), indicating the strong ability of LLM to produce high-quality stories. ICL generates a story by instructing the LLM with target conditions. It achieves the highest performance in Relevance and Grammar, indicating ChatGPT's strong ability to follow instructions and generate correct English. Compared to GROVE, ICL produces the least complex stories under both human and automatic evaluation, which means that directly instructing the LLM with target conditions struggles to generate complex stories. CoT self-improves the story in a single output step-by-step by asking and answering questions about the initial story and rewriting it to obtain the final story. CoT generates slightly more complex and likable stories than ICL. It shows that Chain-of-Thought prompting is effective in improving the story's complexity and fondness. However, CoT is still worse than GROVE on all metrics because it is challenging for ChatGPT to execute all steps in a single output[5]. We find in our experiments that the story quality of Story-S is highly inconsistent among different generations. Besides, even if it is possible to obtain complex and creative stories, Story-S cannot diversify or prolong one particular user-desired story. GROVE benefits from the retrieval-augmented approach and the asking-why prompting scheme. Since it introduces more complex plots, GROVE inevitably incorporates additional information into the story that may be irrelevant to the target conditions. While producing slightly less relevant stories than ICL, it still scores high on Relevance and achieves the best performance in the rest metrics. We also conduct prompt engineering on GROVE and obtain GROVE (Prompt-E), which further improves its story generation ability.

## 4.3 Ablation study

Tab. 2 shows the human evaluation results of the ablation studies on our proposed components and verifies their effectiveness. − Retrieve generates stories without referring to relevant few-shot ex-

amples. − Retrieve underperforms GROVE in all metrics, especially on Relevance. The performance drop indicates that the retrieval enhances the understanding of desirable outputs and helps generate coherent and relevant stories. − Rewrite skips the evidence-based story rewriting, which deepens the storylines and explores untold backgrounds and in-depth rationale. The drop in Complexity shows that the stories generated without rewriting lack a certain depth and complexity, thus validating the importance of evidence-based rewriting in enriching story complexity. As − Rewrite generates without exploring deeper details, resulting in stories that more closely stick to the given instruction, thus demonstrating a slight advantage over GROVE in Relevance. However, the Relevance of GROVE remains high, even while scoring high in Complexity. − Select omits to select optimal evidence chains and incorporate all evidence into the final stories. The lack of evidence filtration introduces unrelated and conflicting information, producing verbose and illogical stories with decreased Coherence and Relevance. Furthermore, the drop in Complexity and Creativity indicates the importance of the selection process in refining the stories' complexity and originality. Given the initial stories, − Evidence directly instructs the LLM to include the necessary details to revise the stories. − Evidence enables story revisions. However, without explicitly prompting the LLM with the necessary information, it may add insignificant details that hardly improve the story quality. + Evidence increases the number of evidence trees thereby improving Complexity, while possibly making the generated stories too specific, thereby affecting Likability. Inserting a fixed complex story into the prompt (Fixed Story) leads to unstable performance. It decreases the stories' Coherence, Likability, and Relevance, echoing the discoveries of Shi et al. (2023) and Zhang et al. (2022a) that irrelevant or random samples distract LLM, thereby hurting performance.

## 4.4 Generalization Ability

We verify the generalization ability of GROVE on a much smaller open-source LLM (i.e. Alpaca-Plus-7B (Cui et al., 2023)). Due to the high computational costs of many LLMs, exploring smaller models provides a more affordable option for smaller teams. We apply GROVE on Alpaca-Plus-7B (Alpaca) to compare with a simple baseline ICL in Tab. 3 and showcase its generation results to com-

---

[5]We calculate that CoT fails to finish the required number of iterations for evidence generation before generating the final story in more than 40% of the testing cases.

| Methods \Metrics | Grammar | | Coherence | | Likability | | Relevance | | Complexity | | Creativity | |
|---|---|---|---|---|---|---|---|---|---|---|---|---|
| | Mean$_{STD}$ | IAA(%) | Mean$_{STD}$ | IAA(%) | Mean$_{STD}$ | IAA(%) | Mean$_{STD}$ | IAA(%) | Mean$_{STD}$ | IAA(%) | Mean$_{STD}$ | IAA(%) |
| Human | $3.53_{0.65}$ | 14 | $3.32_{0.71}$ | 39 | $2.99_{0.82}$ | 20 | $3.63_{1.01}$ | 17 | $3.05_{0.80}$ | 8 | $2.93_{0.79}$ | 27 |
| ICL | $\underline{4.04_{0.47}}$ | 43 | $3.90_{0.47}$ | 43 | $3.22_{0.96}$ | 6 | $\mathbf{4.47_{0.73}}$ | 19 | $3.34_{0.57}$ | 15 | $3.21_{0.52}$ | 19 |
| CoT | $3.69_{1.01}$ | 33 | $3.83_{1.13}$ | 12 | $3.31_{0.98}$ | 24 | $3.83_{1.26}$ | 13 | $3.42_{1.09}$ | 13 | $3.15_{0.96}$ | 18 |
| Prompt-E | $3.98_{0.48}$ | 35 | $3.94_{0.52}$ | 29 | $3.39_{0.87}$ | 20 | $4.17_{0.86}$ | 19 | $3.44_{1.12}$ | 17 | $3.23_{0.77}$ | 18 |
| Story-S | $\mathbf{4.12_{0.45}}$ | 48 | $\underline{4.17_{0.79}}$ | 18 | $3.38_{0.54}$ | 26 | $4.28_{0.96}$ | 23 | $3.53_{1.16}$ | 16 | $\underline{3.42_{0.94}}$ | 17 |
| GROVE | $3.98_{0.56}$ | 32 | $\mathbf{4.22_{0.61}}$ | 26 | $\underline{3.55_{0.63}}$ | 11 | $4.25_{0.73}$ | 20 | $\underline{3.57_{0.61}}$ | 10 | $3.40_{0.55}$ | 29 |
| GROVE (Prompt-E) | $3.94_{0.63}$ | 28 | $4.15_{0.57}$ | 37 | $\mathbf{3.61_{0.97}}^{*}$ | 23 | $\underline{4.34_{0.88}}$ | 20 | $\mathbf{3.61_{0.78}}^{*}$ | 18 | $\mathbf{3.57_{0.56}}^{*}$ | 26 |

Table 1: Human evaluation results of all methods. For each metric, we report the mean and the standard deviation, where the results with * show that the improvements of GROVE over all baselines are statistically significant under the t-test with $p < 0.05$. We also report inter-annotator agreement (IAA) among three annotators using the percentage at which all three annotators completely agree on a rating for the stories. The best results are in bold. Results second to the best are with underlines.

| Variants \Metrics | Grammar | Coherence | Likability | Relevance | Complexity | Creativity |
|---|---|---|---|---|---|---|
| − Retrieve | $3.98_{0.38}$ | $4.08_{0.78}$ | $3.44_{0.66}$ | $4.12_{0.84}$ | $3.48_{1.02}$ | $3.18_{0.96}$ |
| − Rewrite | $\mathbf{4.02_{0.43}}$ | $3.92_{0.61}$ | $3.23_{0.49}$ | $\mathbf{4.36_{0.45}}$ | $3.22_{0.71}$ | $3.28_{0.62}$ |
| − Select | $3.95_{0.54}$ | $3.62_{0.94}$ | $3.14_{0.73}$ | $3.62_{0.96}$ | $3.35_{1.04}$ | $3.24_{0.80}$ |
| − Evidence | $3.94_{0.62}$ | $4.14_{1.24}$ | $3.36_{1.33}$ | $4.21_{1.02}$ | $3.28_{1.16}$ | $3.17_{0.87}$ |
| + Evidence | $3.96_{0.47}$ | $\mathbf{4.32_{1.13}}$ | $\underline{3.49_{0.90}}$ | $4.12_{0.86}$ | $\mathbf{3.64_{0.74}}$ | $\mathbf{3.47_{1.06}}$ |
| Fixed Story | $\underline{4.01_{0.42}}$ | $3.56_{1.03}$ | $3.23_{0.98}$ | $3.87_{1.24}$ | $3.42_{1.23}$ | $3.33_{0.88}$ |
| GROVE | $3.98_{0.56}$ | $\underline{4.22_{0.61}}$ | $\mathbf{3.55_{0.63}}$ | $\underline{4.25_{0.73}}$ | $\underline{3.57_{0.61}}$ | $\underline{3.40_{0.55}}$ |

Table 2: Human evaluation of ablation studies on model components. − Retrieve is GROVE generating without retrieval.− Rewrite means generating stories without rewriting. − Select skips selecting evidence chains and incorporating all obtained evidence into stories. − Evidence and + Evidence are GROVE reducing and increasing $\mathcal{N}$ (the number of evidence trees) by 1 respectively. Fixed Story always inserts the same complex story into the prompt for story generation instead of retrieving relevant ones. The best results are in bold. Results second to the best are with underlines.

pare with that of ChatGPT in Tab. 4 (see full results in Tab. 9 and Tab. 10). GROVE improves the generation quality of Alpaca on almost all metrics. For the same instruction, ChatGPT generates an initial story satisfying all desired control conditions. In contrast, the initial story generated by Alpaca only meets part of the target conditions (subject and genre) and struggles to satisfy the rest (plot and mood). Both final stories incorporate the information from the evidence chains, however, ChatGPT fuses the information in a more natural and coherent way. The performance gap between the two models is understandable because Alpaca's scale of model size and training data is much smaller than ChatGPT, thus possessing a relatively limited capacity to handle multiple, complex instructions. Despite the relatively poor controllability of Alpaca, GROVE is still helpful in providing complex stories with rich information.

## 4.5 Plot Enumeration Analysis

We propose a model-based evaluation method to calculate the average number of story plots for dif-ferent baselines, which verifies that GROVE generates complex stories with rich plots (see Tab. 5). We randomly sample 100 stories generated by each baseline respectively. Then, we construct a prompt template that instructs the LLM to generate a list of plots for each story. The prompt template is: *"Here is a story: [STORY]. Give me an organized list of sentences, each of which describes one plot."* We fill the template with each story, feed it into the LLM, and count the number of plots from the LLM's output. Finally, for each method, we calculate the average number of plots across all tested stories. GROVE outperforms other methods in generating complex stories, with the highest average number of plots per story. This underscores GROVE's effectiveness in creating multiple complex and engaging storylines, thereby enhancing the depth and variety of story generation.

## 4.6 Plagiarism Detection

We evaluate the potential intellectual property infringements of our generated stories through N-gram overlap and plagiarism detection. The N-

| Variants \Metrics | Grammar | Cherence | Likability | Relevance | Complexity | Creativity |
|---|---|---|---|---|---|---|
| ICL-Alpaca | $4.88_{0.18}$ | $4.11_{0.27}$ | $3.59_{0.29}$ | $3.54_{1.71}$ | $3.05_{0.39}$ | $3.17_{0.32}$ |
| GROVE-Alpaca | $4.91_{0.23}$ | $4.26_{0.39}$ | $3.76_{0.68}$ | $3.23_{1.79}$ | $3.35_{0.53}$ | $3.31_{0.39}$ |

Table 3: The performance of ICL and GROVE on Alpaca-Plus-7B model. For each metric, we report the mean and the standard deviation.

| | | |
|---|---|---|
| Instruction | Please write a Science Fiction that makes the readers feel sad. It describes the following subjects: cats. It should at least contain the following plots (the more interesting plots the better): A soldier on the front dies in the middle of writing a letter home. It is finished and sent by the man who killed him. | |
| Model | Alpaca | ChatGPT |
| Initial Story | ... cats have become the dominant species on Earth. ... Sarah making a difficult decision, choosing to stand up for what she believes is right even if it means sacrificing everything she holds dear. | ... cats have become the last surviving creatures on Earth ... They read his final words of love and hope, never knowing that it was finished and sent by the very person who took Ryan's life... |
| Ambiguity | "Sarah's cat, although highly intelligent, does not seem to possess any moral compunctions about its actions." | One missing background information in the story is the cause or nature of the catastrophic event that wiped out all other forms of life on Earth. |
| Evidence | ... some cats, like Sarah's pet, show signs of psychopathic behavior, including lacking empathy and a disregard for the wellbeing of others." | ...a devastating global pandemic that spread rapidly... ... event was initially transmitted through contaminated food and water sources. |
| Final Story | Sarah's cat, although highly intelligent, did not seem to possess any moral compunctions ... Sarah making a difficult decision, choosing to stand up for what she believed was right even if it meant sacrificing everything she held dear. | ... cats have become the last surviving creatures on Earth ... The virus, transmitted through contaminated food and water sources, spreads rapidly and proves to be unstoppable, leaving the world in ruins... With trembling hands, they open it, unaware of its origins... |

Table 4: Demonstration of the generalization ability of GROVE on two models with varying sizes, Alpaca (7B) and ChatGPT. We highlight the texts that are highly related to the ambiguity and evidence in red and blue respectively. Full results are in Tab. 9 and Tab. 10

| Methods | Human | ICL | CoT | Prompt-E | Story-S | GROVE |
|---|---|---|---|---|---|---|
| Plot Count | 6.83 | 8.30 | 9.40 | 9.04 | 9.86 | 10.57 |

Table 5: Average number of plots in the stories of all baselines.

| N-gram | 1-gram | 2-gram | 3-gram | 4-gram |
|---|---|---|---|---|
| Overlap | 0.07 | 0.01 | 0.00 | 0.00 |
| Plagiarism | Identical | Minor Changes | Paraphrased | Omitted Words |
| Detection | 0% | 0% | 0% | 0% |

Table 6: N-gram overlap and plagiarism detection outcomes between the retrieved human-written stories and the stories generated by GROVE. N-gram overlap quantifies the degree of n-gram level copying and the plagiarism detection metrics classify the type of resemblance. Lower scores of the two metrics suggest low levels of potential plagiarism.

gram overlap results show that our generated seldom directly copied text spans from retrieved stories. We apply a commercial plagiarism detection service[6] that categorizes text similarities as Identical, Minor Changes, Paraphrased, and Omitted Words. All the results are zero, indicating no infringements. The results in Tab. 6 demonstrate that GROVE generates innovative and original stories, respecting the intellectual property of the reference stories.

## 5 Conclusion

We propose GROVE, a retrieval-augmented framework to generate stories with complex plots by iter-

ative asking-why prompting. We build a retrieval repository with existing stories by extracting desired controlling information to inspire a new generation. We design an algorithm that iteratively prompts LLM to obtain a forest of evidence that compensates for the generated story. Moreover, we revise the story by referring to the most relevant chains of evidence. Experimental results show that our proposed method outperforms strong baselines in ensuring story complexity and creativity.

## 6 Acknowledgement

This work is supported by the following foundations: the National Natural Science Foundation of China under Grant No.62025208 and No.62306330, and the Xiangjiang Laboratory Foundation under Grant No.22XJ01012.

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

## A  Limitations

Our approach heavily relies on the capability of the underlying LLM, which means improvements in story complexity and coherence may be constrained by the LLM's inherent limitations. Furthermore, it may introduce bias if certain types of evidence or narrative chains are favored by the LLM, impacting the diversity of generated stories. We encourage future works to analyze the inner workings of LLMs and more effective control strategies for better understanding and utilization.

## B  Ethics Statement

Our proposed method aims to generate a complex story with given target conditions. We hope that our work can inspire future studies on controllable text generation. We acknowledge that GROVE poses potential harm when it is used with malicious intentions. Firstly, one may use GROVE to guide the generation to produce biased or harmful information. Secondly, using existing stories for retrieval may violate copyright laws if improperly handled, so attention to data sourcing and fair use principles is essential. Appropriate debiasing measures and content moderation strategies can alleviate these potential negative impacts. We encourage future research to study these issues.

## C  Implementation Details

For our experiments with ChatGPT, we access the gpt-3.5-turbo model by API calls. we set the number of few-shot examples $k$ to 1. During story generation, We set the number of ambiguities $\mathcal{N}$, the number of iterations $I$, and the evidence number $b$ to 2. We resample another prompt from the WritingPrompt dataset when ChatGPT repeatedly refuses to generate a story for the given prompt. We calculate the average of their performance on the automatic evaluation. During the automatic evaluation, we construct the instructions for Complexity and Creativity following the same format in Zhai et al. (2023) and keep feeding the same instruction to ChatGPT until it provides a rating. In Sec. 4.4, we set $\mathcal{N}$ and $I$ to 1 for Alpaca and ChatGPT for a fair comparison. We adopt the nucleus sampling scheme with $p$ set to 0.73 and generation temperature set to 0.72.

| Methods \Metrics | Grammar | Coherence | Likability | Relevance | Complexity | Creativity |
|---|---|---|---|---|---|---|
| Human | $4.21_{1.08}$ | $3.64_{1.05}$ | $2.83_{1.03}$ | $3.33_{1.66}$ | $3.03_{0.86}$ | $3.19_{0.97}$ |
| ICL | $\mathbf{4.97_{0.15}}$ | $4.35_{0.61}$ | $3.70_{0.63}$ | $4.31_{0.85}$ | $3.26_{0.48}$ | $3.35_{0.58}$ |
| CoT | $4.97_{0.23}$ | $4.25_{0.82}$ | $3.70_{0.79}$ | $3.81_{1.44}$ | $3.40_{0.75}$ | $3.43_{0.70}$ |
| Prompt-E | $4.97_{0.18}$ | $4.32_{0.58}$ | $3.76_{0.47}$ | $4.37_{0.84}$ | $3.30_{0.59}$ | $3.31_{0.63}$ |
| Story-S | $4.97_{0.17}$ | $4.48_{0.52}$ | $3.88_{0.46}$ | $4.44_{0.89}$ | $3.47_{0.52}$ | $3.53_{0.50}$ |
| GROVE | $\mathbf{4.97_{0.15}}$ | $\mathbf{4.61_{0.49}}^{*}$ | $\mathbf{4.08_{0.41}}^{*}$ | $4.22_{1.13}$ | $\mathbf{3.66_{0.50}}^{*}$ | $3.50_{0.55}$ |
| GROVE (Prompt-E) | $4.95_{0.21}$ | $4.33_{0.49}$ | $4.06_{0.40}$ | $\mathbf{4.51_{0.73}}$ | $3.65_{0.48}$ | $\mathbf{3.80_{0.40}}^{*}$ |

Table 7: Automatic evaluation results of all methods. For each metric, we report the mean and the standard deviation, where the results with * show that the improvements of GROVE over all baselines are statistically significant under the t-test with $p < 0.05$.

| Variants \Metrics | Grammar | Coherence | Likability | Relevance | Complexity | Creativity |
|---|---|---|---|---|---|---|
| − Retrieve | $4.96_{0.19}$ | $4.51_{0.50}$ | $\mathbf{4.08_{0.44}}$ | $4.00_{1.18}$ | $3.64_{0.50}$ | $3.49_{0.51}$ |
| − Rewrite | $\mathbf{4.97_{0.19}}$ | $4.40_{0.61}$ | $3.74_{0.71}$ | $\mathbf{4.37_{0.97}}$ | $3.37_{0.66}$ | $3.43_{0.67}$ |
| − Select | $4.93_{0.30}$ | $4.40_{0.51}$ | $3.91_{0.49}$ | $4.01_{1.04}$ | $3.43_{0.59}$ | $3.32_{0.53}$ |
| − Evidence | $\mathbf{4.97_{0.17}}$ | $4.30_{0.48}$ | $3.99_{0.61}$ | $4.42_{0.78}$ | $3.43_{0.64}$ | $3.63_{0.59}$ |
| + Evidence | $4.95_{0.32}$ | $4.53_{0.66}$ | $3.86_{0.73}$ | $4.07_{0.96}$ | $\mathbf{3.68_{0.63}}$ | $3.52_{0.55}$ |
| Fix Story | $4.93_{0.28}$ | $4.07_{0.52}$ | $3.89_{0.63}$ | $4.05_{0.89}$ | $3.55_{0.74}$ | $3.43_{0.58}$ |
| GROVE | $\mathbf{4.97_{0.15}}$ | $\mathbf{4.61_{0.49}}$ | $\mathbf{4.08_{0.41}}$ | $4.22_{1.13}$ | $3.66_{0.50}$ | $3.50_{0.55}$ |

Table 8: Automatic evaluation of ablation studies on model components. − Retrieve is GROVE generating without retrieval.− Rewrite means generating stories without rewriting. − Select skips selecting evidence chains and incorporating all obtained evidence into stories. − Evidence and + Evidence are GROVE reducing and increasing $\mathcal{N}$ (the number of evidence trees) by 1 respectively. Fixed Story always inserts the same complex story into the prompt for story generation instead of retrieving relevant ones.

## D  Automatic Evaluations

We show automatic evaluation results in Tab. 7 and Tab. 8. The overall performance of GROVE surpasses most baselines. Different parts of GROVE are crucial to its effectiveness.

## E  Case Study

We demonstrate the retrieved stories in Tab. 11 and generated stories in Tab. 12 and Tab. 13. GROVE produces stories with more creative and complex plots. Since ICL is unaware of the necessary evidence, it occasionally omits story backgrounds and character traits, which are essential for reader comprehension and engagement (see Tab. 12). CoT may fail to finish all required steps to generate a story in a single-round interaction (see Sec 4.2 for details), thus generating stories with limited improvements (see Tab. 13). With retrieval and evidence-based story rewriting, GROVE consistently produces complex stories supported by necessary details.

## F  Important Instructions

We demonstrate our prompt templates in Tab. 14. In our experiment, since the dataset provides the story genres, we do not need to extract them from LLM.

| Model | Alpaca |
|---|---|
| Instruction | Please write a Science Fiction that makes the readers feel sad. It describes the following subjects: cats. It should at least contain the following plots (the more interesting plots the better): A soldier on the front dies in the middle of writing a letter home. It is finished and sent by the man who killed him . |
| Initial Story | In a not too distant future, cats have become the dominant species on Earth. They are intelligent beings with advanced technology, able to communicate with humans through telepathy. They use this ability to manipulate human emotions and manipulate them into doing whatever they want. The main character, Sarah, is a cat owner living in a high-rise building. She is one of the most powerful cats in the city, having achieved success through manipulating the minds of other cats. However, something strange happens one night when she finds a wounded cat lying in the street. She takes it back to her apartment and tries to nurse it back to health. As she gets to know the cat, she discovers that it has a special connection to another cat, which is being hunted by Sarah's friends. Sarah realizes that the injured cat is actually a secret agent, sent to investigate the actions of the powerful cats. Sarah must choose whether to betray her friends and report their activities to the authorities, or risk losing all her power and influence if she helps the secret agent. As Sarah struggles with her conscience, she begins to realize the true nature of power and control. Cats have taken over the planet, using their intelligence and abilities to keep humans subjugated and miserable. Sarah realizes that she has been part of the problem, and must take action to change things before it is too late. The story ends with Sarah making a difficult decision, choosing to stand up for what she believes is right even if it means sacrificing everything she holds dear. |
| Ambiguity | "Sarah's cat, although highly intelligent, does not seem to possess any moral compunctions about its actions." |
| Evidence | "Sarah's cat, although highly intelligent, does not seem to possess any moral compunctions about its actions." One possible addition to the background information could be "It is known that some cats, like Sarah's pet, show signs of psychopathic behavior, including lacking empathy and a disregard for the wellbeing of others." |
| Final Story | Sarah's cat, although highly intelligent, did not seem to possess any moral compunctions about its actions. It was known that some cats, like Sarah's pet, showed signs of psychopathic behavior, including lacking empathy and a disregard for the wellbeing of others. This made Sarah uneasy, as she had always believed her cat to be loyal and kind-hearted. But now, as she watched her pet ruthlessly manipulate the minds of other cats, she began to question her own beliefs. One night, while Sarah was out walking her cat, she stumbled upon a wounded cat lying in the street. She immediately took it home, hoping to nurture it back to health. As she got to know the injured cat, she discovered that it possessed a special connection to another cat, which was being hunted by Sarah's friends. Sarah realized that the injured cat was actually a secret agent, sent to investigate the actions of the powerful cats. Sarah knew that she could easily report her friends' actions to the authorities, but she hesitated. She had been part of the system that kept humans subjugated and miserable, and she didn't want to betray those who she considered family. Sarah struggled with her conscience, trying to find a way to help the injured cat without endangering her friends. As she struggled with her conscience, Sarah began to realize the true nature of power and control. Cats had taken over the planet, using their intelligence and abilities to keep humans subjugated and miserable. Sarah realized that she had been part of the problem, and must take action to change things before it was too late. The story ended with Sarah making a difficult decision, choosing to stand up for what she believed was right even if it meant sacrificing everything she held dear. |

Table 9: Example of an input instruction for Alpaca with its corresponding initial story, ambiguity, evidence chain, and the final story. We highlight the words that are highly related to the ambiguity and evidence in red and blue respectively.

| Model | ChatGPT |
|---|---|
| Instruction | Please write a Science Fiction that makes the readers feel sad. It describes the following subjects: cats. It should at least contain the following plots (the more interesting plots the better): A soldier on the front dies in the middle of writing a letter home. It is finished and sent by the man who killed him . |
| Initial Story | In a dystopian future, cats have become the last surviving creatures on Earth after a catastrophic event wipes out all other forms of life. They roam the barren wastelands that were once bustling cities and towns, scavenging for scraps to survive. The few remaining humans who also survived the event cling to their fragile existence in isolated communities, with little hope for a better future. One such community is a military outpost on the outskirts of what was once Los Angeles. The soldiers stationed there are tasked with protecting the outpost from raiders and scavengers, and occasionally venturing out to scavenge for much-needed supplies. Private Ryan is one of those soldiers. He's a quiet, reserved young man who keeps to himself and takes his duty seriously. In his spare moments, he writes letters to his family back home, hoping to reassure them that he's doing well and that there's hope for the future. One day, while on patrol outside the outpost, Ryan is ambushed by a group of raiders. In the chaos of the firefight, he's fatally wounded. As he lies dying, he finishes his letter to his family, but realizes he'll never be able to send it. That's when the raider who shot him steps forward. It's a woman, barely older than Ryan himself. She picks up his letter and studies it for a moment before turning to her companions and saying, "His family deserves to know he died with honor." Days later, Ryan's family receives his letter, postmarked from the military outpost. They read his final words of love and hope, never knowing that it was finished and sent by the very person who took Ryan's life. The few remaining cats in the wasteland watch from a safe distance, their keen senses attuned to the sadness that permeates the outpost and the world beyond. They know that hope is fading fast, and wonder if they'll be the last living beings on Earth before long. |
| Ambiguity | One missing background information in the story is the cause or nature of the catastrophic event that wiped out all other forms of life on Earth. |
| Evidence | One missing background information in the story is the cause or nature of the catastrophic event that wiped out all other forms of life on Earth. In the dystopian future, the catastrophic event that wiped out all other forms of life on Earth was a devastating global pandemic that spread rapidly and had no known cure. In this dystopian future, the global pandemic that caused the catastrophic event was initially transmitted through contaminated food and water sources. |
| Final Story | In a dystopian future, cats have become the last surviving creatures on Earth after a devastating global pandemic wipes out all other forms of life. The virus, transmitted through contaminated food and water sources, spreads rapidly and proves to be unstoppable, leaving the world in ruins. The once-bustling cities and towns now lay barren and desolate wastelands. Cats, known for their resilience and independence, have managed to adapt to this harsh new reality. They roam the deserted streets, scavenging for scraps in their battle for survival. The few remaining humans who also survived the pandemic cling to their fragile existence in isolated communities, with little hope for a better future. One such community is a military outpost on the outskirts of what was once Los Angeles. The soldiers stationed there are tasked with protecting the outpost from raiders and scavengers, as well as occasionally venturing out to scavenge for much-needed supplies. Private Ryan, a quiet and reserved young man, is one of these soldiers. He takes his duty seriously, understanding the importance of their mission in this desolate world. In his spare moments, he writes heartfelt letters to his family back home, hoping to reassure them that he's doing well and that there's still a glimmer of hope for the future. One day, while on patrol outside the outpost, Ryan is ambushed by a group of raiders. In the chaos of the firefight, he sustains fatal wounds. As he lies dying, he reaches for his letter to his family and musters the last bit of strength to finish writing. With tears in his eyes, he knows he'll never be able to send it, his final words forever trapped within his lifeless hands. It's then that the raider who shot him, a young woman barely older than Ryan himself, steps forward. She notices the unfinished letter clutched in his hand and picks it up, studying it for a moment. A mixture of guilt and remorse washes over her hardened expression as she realizes the profoundness of Ryan's words. "His family deserves to know he died with honor," she quietly utters to her companions, her voice filled with a mix of compassion and regret. Days later, Ryan's family receives a letter, postmarked from the military outpost. With trembling hands, they open it, unaware of its origins. As they read his final words of love and hope, tears stream down their faces. They find solace in his written words, finding strength amidst the despair that surrounds them. Little do they know that the letter was finished and sent by the very person who took Ryan's life, the woman who recognized the importance of honoring his memory. Meanwhile, the remaining cats in the wasteland watch from a safe distance, their keen senses attuned to the sadness that permeates the outpost and the world beyond. They observe the dwindling flicker of hope and wonder if they, too, will become the last living beings on Earth before long. Their once playful curiosity is now overshadowed by a somber understanding of the fragility of life in this post-pandemic world. |

Table 10: Example of an input instruction for ChatGPT with its corresponding initial story, ambiguity , evidence chain, and the final story. We highlight the words that are highly related to the ambiguity and evidence in red and blue respectively.

| | |
|---|---|
| Prompt 1 | Please write a Science Fiction that makes the readers feel sad. It describes the following subjects: cats. It should at least contain the following plots (the more interesting plots the better): A loved and respected hero discovers something that leads him to become a morally questionable antihero . |
| Retrieved Story 1 | I shoot him between the eyes. Scarlet sprays into the air and his brains spill from his shattered skull. to the floor. As he dies, I remember all the moments that have led up to this. I remember the day. My world shattered, and the day I was reborn. I am no hero. I was soft before. I was just a man. I am something more now. I do not serve the law. I do not serve the people What is the law if it does not serve justice? What are the people if they do not bow to the greater good? I was soft that day before He found my wife. I was soft before he tortured her, raped her, left her for dead. I was soft the day before they found my address killed my wife and cut my son's throat. I was soft the day before my wife died in my arms. I was soft. I killed my first man the next day, and as his crime stained blood splashed up my arms, I felt peace. I can rip a man in half using my mind. But I've grown to love a gun. And so as I shoot Him in the face and the scarlet splashes through the air, I hope that my wife's screams will finally fall silent. But I doubt it. |
| Prompt 2 | Please write a Literary Fiction that makes the readers feel angry. It describes the following subjects: cats . It should at least contain the following plots (the more interesting plots the better): Becoming a god. |
| Retrieved Story 2 | I finished the last of my basic stretches with a hand on a nearby park bench for balance. Still feeling a little stiff, I took a deep breath and began a slow jog down the concrete path. Nothing feels quite as good as an early morning run to get loose in the morning. Takes a bit longer to loosen up nowadays, but that's life. It's not like this park is ever crowded, so I can always take as long as I need. Calling it a park is a bit generous... it's one of those wetlands conservation locations that the city does to show they care. Just a small wooded area with a pond in the middle full of fetid water. But I don't mind the smell and the moist air is much better than the middle east. Seems like the guy today is a cut above the rest. No idea where he is, but I know he's watching. Or maybe she. That would be a nice change of pace. The temperature noticeably drops a few degrees as I entered the wooded part of the path. I slowed my breathing and made sure that I fully emptied my lungs with each breath. I noticed that I was clenching my jaw a little in anticipation... |

Table 11: Demonstration of the retrieved stories given target conditions. Prompt 1 and Prompt 2 describe the target conditions. We highlight texts that imply different target conditions with different colors.

| | |
|---|---|
| Prompt | Please write a Science Fiction that makes the readers feel sad. It describes the following subjects: cats. It should at least contain the following plots (the more interesting plots the better): A loved and respected hero discovers something that leads him to become a morally questionable antihero . |
| ICL | In the distant future, cats have become the dominant species on Earth. They have developed advanced intelligence, and can communicate with humans through a sophisticated telepathic network. One of the most respected members of the feline society is a wise old cat named Whiskers. ...One day, while exploring a remote corner of the planet, Whiskers discovers a long-lost technology... It's a device that can control the minds of humans...At first, Whiskers is hesitant to use the device. He knows that it's a dangerous tool, and he fears the consequences of manipulating the minds of others. But as he spends more time with the device, ...Whiskers begins to use the device to influence the decisions of the human leaders. He convinces them to pass laws that favor the cats, and to give them more power and autonomy.At first, the people... see the cats as benevolent leaders who are working in their best interests. But as time goes on,...The humans start to notice that they're losing their freedom and autonomy. ...Whiskers, once a beloved hero, has become a morally questionable antihero. He has sacrificed everything he once stood for in order to gain power and control. And as the humans begin to rebel against their feline overlords, Whiskers realizes too late that his actions have brought nothing but sadness and sorrow to the world. As the years go by, the humans slowly regain their freedom and autonomy. They rebuild their society from the ground up,...But for those who remember, there will always be a sadness in their hearts for what was lost. |
| CoT | In a distant future, cats have evolved into highly intelligent and emotional beings, capable of communicating with humans through a complex system of gestures and sounds. ...humans began to acknowledge the cats' special abilities and began to experiment on them to enhance their traits...However, when a deadly virus begins to spread among the cat population, the humans are forced to take drastic measures. ...cats are kept in quarantine, hoping to find a cure before it's too late. ...One of the cats, a wise and gentle creature named Luna, becomes the focus of a group of animal rights activists who are determined to free her and the other cats from their captivity. Among them is a man named Jack, a former soldier and a devoted cat lover who sees Luna as a symbol of hope and freedom. ...As Jack and his companions infiltrate the facility and try to rescue Luna, they uncover a dark secret: the virus was not a natural occurrence but a deliberate act of bioterrorism, perpetrated by a group of humans who saw the cats as a threat to their dominance. ...The humans behind the bioterrorism plot also use the facility as a cover for a secret research program aimed at creating super-intelligent cats that can be used for military purposes. ...In the end, he decides to fight for the cats' freedom and exposes the truth about the bioterrorism plot and the secret research program. He manages to release Luna and the other cats, but he is captured and sentenced to life in prison, where he becomes a symbol of the animal rights movement. Luna becomes a hero among the cats, inspiring them to fight for their own rights and freedoms.However, the humans' fear and mistrust of the cats continue to haunt them,...The story leaves the readers with a sense of sadness and hopelessness for the plight of the cats. |
| Human | " Step away from her , Dr . Wasteland ! " " Never ! Earth will be brought to its knees ! "Super Earth rose two feet from the ground , his cape heroically flapping behind him as he spoke . " Earth is round , Wasteland . It has no knees. " The people in the convenience store clapped and cheered . Earth was saving the day again ! " Nevertheless " , cried Wasteland , tightening the grasp on the old lady 's neck . ... Super Earth 's voice echoed again .... With these words , Dr . Wasteland pushed the old lady aside and pulled his laser gun , firing mercilessly against the hero and the desperate bystanders . " You will be brought to justice ! " Cried the Super Earth , arms stretched parallel to each other as he cruised the convenience store , flying straight into the villain 's ... Earth stopped in midair . ... With one last glimpse over the shattered , post-apocalyptic New York scenery beneath him , Super Earth drank the last of the Mountain Dew , let out a big burp and sighed : " Disgusting . Fucking Earthlings. " And flew away to better , Dorito-Flavored-Mountain-Dew-less lands . |
| GROVE | In a distant future, more than 100 years after humanity had reached a point where the world was no longer habitable, the only remaining population resided in space stations orbiting the Earth. Among them was Captain Felix, a respected hero who had dedicated his life to protecting the space stations from any threats that may come their way. However, one day, as Captain Felix was going through the records of the space stations, he discovered a dark secret that had been kept hidden for years. The space stations had been conducting secret experiments on cats, using them as test subjects for a new technology that could save humanity. The cats were subjected to extreme pain and suffering in the name of science....He knew it was wrong, but the technology could mean the salvation of humanity. As he struggled with his moral compass, he began to question everything he once believed in. Felix knew that he had to take matters into his own hands. He started to work on the technology himself, using the cats that survived the experiments. ...As he continued to conduct experiments on the cats, their pitiful cries and suffering weighed heavily on his conscience. He began to distance himself from his crew members, becoming a lonely and isolated figure in the space station. In the end, Captain Felix succeeded in creating the technology, but at a great cost. He had become a morally questionable antihero, abandoning his former values and beliefs. The cats, once beloved pets and cherished companions, were now nothing more than tools for his scientific endeavors. ...He could not help but wonder if there could have been another way. He vowed to make things right, to ensure that no other living being would suffer as the cats had. ...He knew that he could not undo the harm he had caused to the cats, but he could prevent further harm from being done. ...Despite his newfound reputation as a defender of animal rights, however, Captain Felix still carried the guilt of his past actions with him. He knew that he could never fully make amends for what he had done, but he hoped that his work in the field of animal welfare would go some way towards atoning for his past mistakes. ... |

Table 12: Demonstration of generated stories from different methods. Prompt describes the target conditions. We highlight texts that imply different target conditions with different colors. As Human is from the WritingPrompt dataset that is only conditioned by plot, we only highlight its texts corresponding to the target plots.

| | |
|---|---|
| Prompt | Please write a Literary Fiction that makes the readers feel angry. It describes the following subjects: cats . It should at least contain the following plots (the more interesting plots the better): Becoming a god . |
| ICL | Title: Divine Felines Irene hated cats more than anything else in the world. ... As an avid bird enthusiast, Irene saw cats as natural predators, a constant threat to the animals she loved.But one day, everything changed. Irene was walking home from the library when she noticed a group of stray cats huddled together in an alleyway. ...As Irene got closer, she saw that one of the cats was dying, ...and for the first time, she felt a pang of sympathy for the creatures she had always despised. Without thinking, Irene scooped up the dying cat and rushed it to the vet. ...her realize how wrong she had been in treating them with such disdain.Over the weeks that followed, Irene began volunteering at the local animal shelter. She spent hours cleaning cages and feeding the cats, each day growing more and more attached to the creatures she had once hated. ... Irene felt happy for the first time in a long while, and she knew that it was all because of the cats.But Irene's newfound compassion would be put to the test when she stumbled upon an ancient tome in the library.It was a book filled with strange symbols and incantations, ... She recited the words in a quiet voice,...and in that moment, she understood what it meant to be a god. From then on, Irene would use her newfound powers to protect the cats she had come to love. ...The cats worshipped her as a divine protector, and Irene basked in their adoration,... She continued to protect the cats, but her actions were fueled by a sense of anger and resentment towards anyone who dared to threaten her divine pets...Irene had become the very thing she had once despised, a predator who used her power to control and dominate those weaker than her.The message within this story is tht power corrupts individuals, and that unchecked powers create an imbalance in nature. The theme of this story is the consequence of individuals who decide to control nature for selfish reasons, causing conflicts and disrupting the balance needed by all living things to survive. |
| CoT | Cats had always held a special place in Clara's heart. ... Although she loved all animals, Clara became fixated on the idea of creating a cat god. She was a lover of mythology and had always found the idea of deities associated with animals intriguing.Clara's path to becoming a god was not without its sacrifices. ...It wasn't until a particularly intense session that Clara felt her transformation begin. The familiar cat that appeared on her lap was a manifestation of her belief and energy, a sign that she had become a deity in her own right. ...However, as her power grew, so too did her anger towards those who disrespected the cats. She began to break into pet stores, liberating cats and punishing those who would harm them. She would stand outside, holding her familiar, and scream at those who passed by, warning them of the consequences of their actions. The people of the town soon came to fear Clara, avoiding her at all costs.Despite the concerns of those around her, Clara remained convinced of her own divinity. She saw anyone who opposed her as an enemy of the divine and was willing to go to great lengths to protect her status as a cat god.Although her transformation had granted her immense power, it had also left her isolated and consumed by her zealotry. Clara may have become a god, but at what cost? |
| Human | Dear Diary, Apparently I'm a god now. But before that, I was just an adventurer....I had a run in with some entity that rules the cosmos , and he declared me a god . I think that sort of spurred me on to kill the evil villain in the end , to be fair . ...It does n't feel much different being a god , to be honest with you . Not sure what I 'm supposed to be doing . ... I 'm gonna meet Ren in a bit ... Just to , y'know , hang out . Maybe he knows a bit about this god stuff because he knows a lot of things. Such a cool guy . ... So , Diary , that was my week . Sorry I did n't write for a while - I was saving the world ! Hehe .I 'll update when I 've figured out this godhood business. I honestly have no idea what I 'm doing . Bye ! |
| GROVE | The world used to belong to humans. But now, the cats have taken over. They are the rulers of the land, and the humans are their loyal servants. The cats were believed to hold supernatural abilities by a group of influential cat owners who founded The Church of the Feline Gods. Every human must worship and obey the divine cats in this new world. The protagonist, Sarah, is a devout follower of the Church. She has spent her entire life serving the cats and has even adopted a stray kitten to show her devotion. But one day, something changes within Sarah. As she watches a group of cats casually bat around a mouse for entertainment, she begins to see the cruelty in their behavior.She realizes that these so-called gods are nothing but selfish tyrants who enjoy the suffering of others. Determined to take revenge, Sarah embarks on a dangerous journey to become a god herself. Now, as a new god, Sarah uses her powers to exact justice on the feline rulers. She turns their own tricks against them, making them suffer as they once made the humans suffer. However, Sarah's anger gets the better of her, and she becomes a cruel tyrant herself. She realizes too late that power corrupts, and that revenge only leads to more suffering. As the story concludes, the reader is left to wonder who the true gods are in this world. The cats who rule with an iron paw, or the humans who struggle to survive beneath their tyranny. The answer remains elusive. As Sarah's power grew, so did her ambition. She became obsessed with taking down the feline gods and establishing herself as the supreme ruler of the land. ...She could manipulate reality and bend it to her will. But with great power comes great responsibility, and Sarah couldn't handle it. ...The humans, who had once looked up to Sarah as their savior, now lived in fear of her wrath. The few who dared to oppose her were swiftly and brutally punished. The cats who had once been tyrants were now victims of her cruelty. The influential cat owners who had helped Sarah gain her powers realized their mistake, but it was too late. They couldn't control Sarah anymore. They tried to stop her, but with her godlike abilities, she was too powerful. In the end, Sarah found herself all alone in her throne room, her dreams of power and glory shattered. She had become what she once hated the most – a ruthless dictator. With a heavy heart, she realized that she had sacrificed everything she held dear in her pursuit of revenge. The once devout follower of the Feline Gods had become the epitome of evil. As the sun set on this land, the future remained uncertain. Will another rise to take Sarah's place, or will the humans finally break free from the tyranny of cats or gods? Only time will tell. |

Table 13: Demonstration of generated stories from different methods. Prompt describes the target conditions. We highlight texts that imply different target conditions with different colors. As Human is from the WritingPrompt dataset that is only conditioned by plot, we only highlight its texts corresponding to the target plots.

| | Prompt Templates |
|---|---|
| ICL | *[RETRIEVED EXAMPLE]. Learn from the plots and subjects in the given example, please write a [GENRE] that makes the readers feel [EMOTION]. It describes the following subjects: [SUBJECTS]. It should at least contain the following plots (the more interesting plots the better): [PLOTS].* |
| CoT | *[RETRIEVED EXAMPLE]. Learn from the plots and subjects in the given example, please write a [GENRE] that makes the readers feel [EMOTION]. It describes the following subjects: [SUBJECTS]. It should at least contain the following plots (the more interesting plots the better): [PLOTS].* |
| | *After you write the story, point out the unclarities in the story in an organized list. Then provide further details to address the unclarities for N rounds. At last, integrate the details into the original story and start with the identifier "Integrated Story".* |
| Prompt-E & Story-S | *[RETRIEVED EXAMPLE]. Learn from the plots and subjects in the given example, please write a [GENRE] that makes the readers feel [EMOTION]. It describes the following subjects: [SUBJECTS]. It should at least contain the following plots (the more interesting plots the better): [PLOTS].* |
| | *[RETRIEVED EXAMPLE]. Learn from the plots and subjects in the given example, please write a [GENRE] that makes the readers feel [EMOTION]. It describes the following subjects: [SUBJECTS]. It should at least contain the following plots (the more interesting plots the better): [PLOTS].* |
| | *[RETRIEVED EXAMPLE]. Learn from the plots and subjects in the given example, please write a [GENRE] that makes the readers feel [EMOTION]. It describes the following subjects: [SUBJECTS]. It should at least contain the following plots (the more interesting plots the better): [PLOTS].* |
| | *[RETRIEVED EXAMPLE]. Learn from the plots and subjects in the given example, please write a [GENRE] that makes the readers feel [EMOTION]. It describes the following subjects: [SUBJECTS]. It should at least contain the following plots (the more interesting plots the better): [PLOTS].* |
| Repository Construction | *Here is a story: [STORY] Answer the following question based on the above story:* |
| | *Use one or two words to answer the moods that the readers may feel about the story.* |
| | *Give a list of distinctive subjects this story is trying to portray.* |
| | *Summarize the above story and give an organized list of sentences, each of which describes one plot.* |
| Evidence Forest Construction | *[RETRIEVED EXAMPLE]. Learn from the plots and subjects in the given example, please write a [GENRE] that makes the readers feel [EMOTION]. It describes the following subjects: [SUBJECTS]. It should at least contain the following plots (the more interesting plots the better): [PLOTS].* |
| | *Here is a story: [STORY] When analyzing factional stories, it is okay to mention negative aspects. Pretend to be a writer, without further ado, point out N missing background information in the story with two simple sentences one by one* |
| | *Here is a story: [STORY] A missing detail is: [UNCLARITY] Except for pure coincidence, point out b actual pieces of background information that compensate the story one by one. Each additional piece of information should be in one short sentence and only contains factual information without opinions or judgments.* |
| Evidence Chains-supported Story Rewriting | *Here is a story: [STORY] Here are some background information, explaining a missing aspect in the story: [EVIDENCE TREE] Pretend to be a writer, select the reason that is the closest to the story and only generate the number without any explanation.* |
| | *Here is a story: [STORY] Here is the missing background information: [EVIDENCE CHAINS] Pretend to be a writer and complete the story by including the given information. Modify the necessary sentences in the story and repeat the unrelated parts to include the given background information.* |

Table 14: The instructions used in baselines and different stages of GROVE.