# OpenReview forum: "GROVE: A Retrieval-augmented Complex Story Generation Framework with A Forest of Evidence"
_EMNLP/2023/Conference — EMNLP 2023 Findings_

### Official Review · Reviewer_C2GE · 2023-07-21

**Typos Grammar Style And Presentation Improvements:** 1. In your tables, show length of the…
**Soundness:** 4

**Excitement:**

3: Ambivalent: It has merits (e.g., it reports state-of-the-art results, the idea is nice), but there are key weaknesses (e.g., it describes incremental work), and it can significantly benefit from another round of revision. However, I won't object to accepting it if my co-reviewers champion it.

**Missing References:**

This work is remotely related
https://arxiv.org/pdf/2212.10012.pdf

**Paper Topic And Main Contributions:**

The paper proposes a story generation method (see its Figure 1). The method generates the stories uses the following steps.
1. classify the attributes of some human written stories using LLM
2. given a target condition, select the human-written stories with the most similar attributes
3. generate an initial story using LLM
4 and 5. keep asking LLM why to generate trees containing some background information or evidence
6. use LLM to select the most relevant information paths from the trees.
7 and 8. revise the initial story using LLM

The experimental results in Table 2 seem to show that the retrieval step (1&2) slightly improves coherence and relevancy, the selection and rewriting (4-8) improves coherence, likability, complexity, creativity, while decreasing the relevancy.

**Reasons To Accept:**

1. It shows that retrieving the stories, asking why, and selecting the proper generated evidence could improve the quality of generated stories.
2. All the methods make sense.
3. The experiments can support its main claims. The evaluation is done by experts with a Master's degrees in English Literature.

**Reasons To Reject:**

Before I start to write my reasons, I would like to first admit that this is my first time to review LLM papers, so I am not familiar with the acceptance standard in the LLM papers. In short, I believe that proposing a complex way to prompt the LLM and show the improvement is not enough. I think you need to show that other simpler or popular approaches cannot lead to similar improvement in order to justify the usage of your complex method.

My main complaint to this paper is its experiments. I discuss its experiment limitations from the following angles.

1. This paper has similar limitations of other LLM papers. The proposed methods are not very novel. It seems to try to make the proposed methods more complex to increase the novelty, but it is unknown if this complex way of prompting is necessary (see the next point). I think using Alpaca-Plus-7B/ChatGPT for generation and ChatGPT for evaluation is not very costly. You should be able to try more baselines and conduct more fine-grained ablation study to verify the effectiveness of the propose approach. For example, you can see how the story quality change as you increase the number of evidence trees. You can try a baseline that always inserts a fixed movie summary from the IMDB dataset into your prompt to show if your retrieval method is really useful.

2. The proposed method is complicated and the improvement of each component is not very large. I wonder if there is some simpler baselines that can achieve similar improvements. I list some examples below.
a) In the appendix tables, I observe that the final stories are usually longer. Maybe we actually don't need to use this complex evidence forest. We can just ask the LLM to revise the initial story to add more details or make it more coherent.
b) In Table 1, ICL seems to give us simpler but more relevant stories. Then, could we simply do some prompt engineering to encourage the LLM to generate more complex stories? Or showing one story with more complex plot as an example and encourage the LLM to write something like that.
c) Table 1 seems to suggest that the human evaluation is highly correlated with the automatic evaluation. Then, I think one simple baseline is to generate several stories and use the automatic evaluation metric to select the best one. It would be good to compare this simple method with GROVE in terms of generation quality and diversity.

3. The experiment is not very comprehensive. All the quantitative results are done on ChatGPT. Not sure whether it is also applicable to other LLMs. Furthermore, in my opinion, ablation study in Table 2 is more important than Table 1 because Table 2 supports the main claim of the paper (retrieval and asking why are effective) and the methods in Table 1 are not very comparable. However, the author does not conduct human evaluation in Table 2.

**Reproducibility:**

3: Could reproduce the results with some difficulty. The settings of parameters are underspecified or subjectively determined; the training/evaluation data are not widely available.

**Reviewer Confidence:**

2: Willing to defend my evaluation, but it is fairly likely that I missed some details, didn't understand some central points, or can't be sure about the novelty of the work.

---

> ### Author Rebuttal · Authors · 2023-08-29
>
> ### Comment
>
> Thank you for your thoughtful and valuable feedback! We follow all your suggestions and we believe we have solved all your concerns as follows! We are pleased to see you may reassess the soundness of our work, based on our new experimental results.
>
> ### More fine-grained ablation studies and human evaluations.
>
> We tried two more ablation studies and conducted human evaluations according to your suggestion. We found that: (1) The human evaluations of $-$ Retrieve, $-$ Rewrite, and $-$ Select align with their performance on automatic evaluation. (2) Inserting a fixed complex story into the prompt (**Fixed Complex Example**) leads to unstable performance and decreases the stories’ Coherence, Likability, and Relevance. (3) Increasing the number of evidence trees (**+ Evidence**) improves Complexity, while possibly making the generated stories too specific, thereby affecting Likability.
>
> #### Table 1: Human evaluation of ablation study.
>
> | Methods\Metrics    | Grammar       | Cherence| Likability    | Relevance     | Complexity    | Creativity    | Length |
> | --------------- | ------------- | ------------- | ------------- | ------------- | ------------- | ------------- | ------ |
> | - Retrieve      | 3.98$_{0.38}$ | 4.08$_{0.78}$ | 3.44$_{0.66}$ | 4.12$_{0.84}$ | 3.48$_{1.02}$ | 3.18$_{0.96}$ | 852.92 |
> | - Rewrite       | 4.02$_{0.43}$ | 3.92$_{0.61}$ | 3.23$_{0.49}$ | 4.36$_{0.45}$ | 3.22$_{0.71}$ | 3.28$_{0.62}$ | 387.74 |
> | - Select        | 3.95$_{0.54}$ | 3.62$_{0.94}$ | 3.14$_{0.73}$ | 3.62$_{0.96}$ | 3.35$_{1.04}$ | 3.24$_{0.80}$ | 740.52 |
> | Fixed Complex Example   | 4.01$_{0.42}$ | 3.56$_{1.03}$ | 3.23$_{0.98}$ | 3.87$_{1.24}$ | 3.42$_{1.23}$ | 3.33$_{0.88}$ | 765.28 |
> | + Evidence      | 3.96$_{0.47}$ | 4.32$_{1.13}$ | 3.49$_{0.90}$ | 4.12$_{0.86}$ | 3.64$_{0.74}$ | 3.47$_{1.06}$ | 754.66 |
> | GROVE           | 3.98$_{0.56}$ | 4.22$_{0.61}$ | 3.55$_{0.63}$ | 4.25$_{0.73}$ | 3.57$_{0.61}$ | 3.40$_{0.55}$ | 712.99 |
>
> We implement **Fixed Complex Example** by always inserting the same complex story into the prompt for story generation instead of retrieving relevant ones. **+ Evidence** is GROVE increasing $\mathcal{N}$ (the number of evidence trees) by 1.
>
> Many studies in story generation claimed that retrieval$^{[1]}$ and in-context demonstration$^{[2]}$ help generate better stories. The performance drop of **Fixed Complex Example** echoes the discoveries of Shi et al.$^{[3]}$ and Zhang et al.$^{[4]}$ that irrelevant or random samples distract LLM, thereby hurting performance.
>
> ### Compare with simpler baselines
>
> We evaluate GROVE on more baselines as you suggested and report the evaluation results in Table 2. We found that **GROVE outperforms baselines on most metrics**.
>
> #### Table 2: Automatic evaluation of more baselines.
>
> | Methods\Metrics    | Grammar       | Cherence| Likability    | Relevance     | Complexity    | Creativity    | Length |
> | ------------------ | ------------- | ------------- | ------------- | ------------- | ------------- | ------------- | ------ |
> | Prompt Engineering | 4.97$_{0.18}$ | 4.32$_{0.58}$ | 3.76$_{0.47}$ | 4.37$_{0.84}$ | 3.30$_{0.59}$ | 3.31$_{0.63}$ | 544.42 |
> | CoT                | 4.97$_{0.23}$ | 4.25$_{0.82}$ | 3.70$_{0.79}$ | 3.81$_{1.44}$ | 3.40$_{0.75}$ | 3.43$_{0.70}$ | 369.79 |
> | Ensemble Learning  | 4.98$_{0.15}$ | 4.40$_{0.74}$ | 3.86$_{0.89}$ | 4.44$_{0.76}$ | 3.41$_{0.44}$ | 3.38$_{0.54}$ | 599.87 |
> | GROVE              | 4.97$_{0.15}$ | 4.61$_{0.49}$ | 4.08$_{0.41}$ | 4.22$_{1.13}$ | 3.66$_{0.50}$ | 3.50$_{0.55}$ | 712.99 |
>
> To clarify, we **have already** used prompts with explicit instruction (i.e. “*the more interesting plots the better*”) to instruct the LLM to generate more plots for all our baselines. The Instruction in Table 3 of our submitted paper demonstrates an exemplary prompt to generate a story. The full prompt template is ***Prompt1***:
>
> >  [RETRIEVED EXAMPLE]. Learn from the plots and subjects in the given example, please write a [GENRE] that makes the readers feel [EMOTION]. It describes the following subjects: [SUBJECTS]. It should at least contain the following plots (the more interesting plots the better): [PLOTS].
>
> We used the same prompt template to generate stories for ICL, **CoT**, and the initial stories in GROVE for a fair comparison.
>
> The **CoT** baseline in our paper **has already** been implemented to instruct the LLM to identify the missing information and iteratively revise the story to include missing backgrounds. We have briefly introduced our implementation of **CoT** in Line 480 of our submitted paper. **CoT** first generates an initial story using ***Prompt1***. Then, it queries the LLM to revise the generated story step-by-step.  The prompt template is ***Prompt2***, where $\mathcal{N}$ is 2 in our experiments:
>
> > After you write the story, point out the unclarities in the story in an organized list. Then provide further details to address the unclarities for $\mathcal{N}$ rounds. At last, integrate the details into the original story and start with the identifier "Integrated Story".
>
> We will list the prompts involved in our baselines and introduce the experimental setting in detail in our revised paper to avoid misunderstandings.
>
> To compare GROVE with more baselines, we implement prompt engineering and ensemble learning and report the experimental results.
>
> **Prompt Engineering** instructs the LLM with different 4 prompts that explicitly say: “Generate a complex and creative story”, “generate a detailed and complex story”, “Ensure that the story is creative and rich in plots.”, and “Generate a long and complex story”, respectively. We calculate the average of their performance on the automatic evaluation.
>
> **Ensemble Learning** uses all 4 instructions in **Prompt Engineering** per generation, ranks the generated stories according to their average scores on all metrics, and selects the highest-ranking story as the final output.
>
> ### Is GROVE also applicable to other LLMs?
>
> Thank you for your suggestion! Following your suggestion, we **quantify** the performance of GROVE on another LLM, Alpaca-Plus-7B (see Table 3). GROVE-Alpaca outperforms ICL-Alpaca on creativity and complexity, verifying GROVE’s generalization ability.
>
> #### Table 3: Automatic evaluation of GROVE on Alpaca-Plus-7B.
>
> | Methods\Metrics    | Grammar       | Cherence| Likability    | Relevance     | Complexity    | Creativity    | Length |
> | --------------- | ------------- | ------------- | ------------- | ------------- | ------------- | ------------- | ------ |
> | ICL-Alpaca      | 4.88$_{0.18}$ | 4.11$_{0.27}$ | 3.59$_{0.29}$ | 3.54$_{1.71}$ | 3.05$_{0.39}$ | 3.17$_{0.32}$ | 370.50 |
> | GROVE-Alpaca    | 4.91$_{0.23}$ | 4.26$_{0.39}$ | 3.76$_{0.68}$ | 3.23$_{1.79}$ | 3.35$_{0.53}$ | 3.31$_{0.39}$ | 396.74 |
>
> It is commonly believed that larger instruction-following LLMs (e.g., ChatGPT) tend to perform better than smaller counterparts (e.g. Alpaca-7B) in performing tasks$^{[5,6]}$. As GROVE improves the performance of Alpaca-Plus-7B and gpt-3.5-turbo, it is reasonable to presume that GROVE can generalize to LLMs with different sizes.
>
> ### Are retrieved stories similar to the initial story?
>
> Yes, the retrieved stories are **similar to the initial stories in terms of the world settings, character designs, and character motivations**. For example, in one of our tested cases, the retrieved story happens in a magic world. The LLM learns from its world setting and produces a story with magic creatures.
>
> Due to the time limit, we are unable to evaluate the similarity between the retrieved stories and the initial story more thoroughly. We will further conduct a detailed human evaluation to analyze the similarity between retrieved and generated stories in multiple aspects. We will also show the retrieved stories in our revised paper.
>
> ### Report the length of stories
>
> Following your suggestion, we report the average lengths of the generated stories of each baseline in Table 4.
>
> Please note that **a longer generation does not necessarily make a better story**. Yang et al. (2022)$^{[7]}$ found that models often generate unnecessarily detailed descriptions that hurt the story quality. Coetzee (2023)$^{8}$ wrote a full-length fiction with GPT-4 and described that some overly detailed chapters “felt like a slog”. Besides, Coetzee also reported that “a consistent level of detail could help”, echoing the motivation of our work. GROVE avoids over-expansion by iteratively asking for the important story backgrounds and revising the initial story.
>
> #### Table 4: Story Lengths of different methods
>
> | Methods | Human  | ICL    | CoT    | $-$ Retrieve | $-$ Rewrite | $-$ Select | GRACE|
> | ------- | ------ | ------ | ------ | ------------ | ----------- | ---------- | ------ |
> | Length  | 705.51 | 401.16 | 369.79 | 852.92       | 387.74      | 740.52     | 712.99 |
>
>
> ### Details of human baseline
>
> In the WritingPrompt dataset, each story is paired with a prompt that describes the requirements for the story. We have sampled stories with their corresponding prompts from the WritingPrompt dataset and evaluated **Relevance** by considering whether the human-authored stories satisfy the requirement of their corresponding prompts.
>
> ### Notations and missing reference
>
> We will add the math notations to Fig 1 and cite the missing reference in the revised paper to improve its understandability.
>
> ### References
>
> [1] Peng Xu, Mostofa Patwary, Mohammad Shoeybi, Raul Puri, Pascale Fung, Anima Anandkumar, and Bryan Catanzaro. 2020. MEGATRON-CNTRL: Controllable Story Generation with External Knowledge Using Large-Scale Language Models. EMNLP
>
> [2] Albert Lu, Hongxin Zhang, Yanzhe Zhang, Xuezhi Wang, and Diyi Yang. 2023. Bounding the Capabilities of Large Language Models in Open Text Generation with Prompt Constraints. Findings of EACL
>
> [3] Shi, F., Chen, X., Misra, K., Scales, N., Dohan, D., Chi, E.H., Schärli, N. & Zhou, D. 2023. Large Language Models Can Be Easily Distracted by Irrelevant Context. ICML.
>
> [4] Yiming Zhang, Shi Feng, and Chenhao Tan. 2022. Active Example Selection for In-Context Learning. EMNLP
>
> [5] Sewon Min, Xinxi Lyu, Ari Holtzman, Mikel Artetxe, Mike Lewis, Hannaneh Hajishirzi, and Luke Zettlemoyer. 2022. Rethinking the Role of Demonstrations: What Makes In-Context Learning Work?. EMNLP.
>
> [6] Wei, J., Tay, Y., Bommasani, R., Raffel, C., Zoph, B., Borgeaud, S., Yogatama, D., Bosma, M., Zhou, D., Metzler, D. and Chi, E.H., 2022. Emergent abilities of large language models. *TMLR*.
>
> [7] Kevin Yang, Dan Klein, Nanyun Peng, and Yuandong Tian. 2023. DOC: Improving Long Story Coherence With Detailed Outline Control. ACL
>
> [8] Chiara Coetzee. 2023. Generating a full-length work of fiction with gpt-4. Medium.

---

### Official Review · Reviewer_ALpk · 2023-08-05

**Soundness:** 2

**Excitement:**

3: Ambivalent: It has merits (e.g., it reports state-of-the-art results, the idea is nice), but there are key weaknesses (e.g., it describes incremental work), and it can significantly benefit from another round of revision. However, I won't object to accepting it if my co-reviewers champion it.

**Paper Topic And Main Contributions:**

The author prposed a retrieve-augmented story generation with the support of evidence forest to fill the missing background information of the story.

The story generation task in this paper is that given the genre, plot, mood, and a subject, the gpt-3.5-turbo model is ask to generate a story. In the proposed method, the first step is to retrieve similary plots from the IMDB plots to enhence the story generation. After obtaining the initial story, the second step is to prompt the model for missing background information (ambiguity). Based on the model output, the next step is to prompt the model to proposed evidences to fill the missing background. Finally, the initial story, the ambiguity, and the new evidences are used to rewrite the story as the final story.

Result show that proposed method achieves higher scores on likability, coherence, complexity, and creativity evaluated by humans and the LLM.

The contribution of this paper is that proposed pipeline helps to obtain more information to improve the initial story and achieves higher scores on several metrics.


**Reasons To Accept:**

The proposed method provides more information to the users on knowing what can be improved from the initial story where it might help humans or machines to (re)-write for a better story.


**Reasons To Reject:**

The whole pipeline is very complex. If we know that the missing background is important for story rewriting, can we prompt the model to iterate on the missing backgrounds in CoT setting.

I believe some performance improvement of the baseline models can be done by simply adding explicit instructions, e.g., “generate a long/creative/complex story”.

For the CoT result, it seems like the problem is that the model failed to finish the story. I am wondering if this can be solved by the 16k model?

It’s unclear what’s the prompt for CoT baseline. It will be more useful to have a full log for each method.


**Reproducibility:**

3: Could reproduce the results with some difficulty. The settings of parameters are underspecified or subjectively determined; the training/evaluation data are not widely available.

**Reviewer Confidence:**

3: Pretty sure, but there's a chance I missed something. Although I have a good feel for this area in general, I did not carefully check the paper's details, e.g., the math, experimental design, or novelty.

**Typos Grammar Style And Presentation Improvements:**

The naming of the proposed methods is some what confusing. Why is the first prompt of asking for missing background called “ask why”? The evidence chain/forest is also not very intuitive to me, since there’s no evidence from other resources but the model generated backgrounds.

---

> ### Author Rebuttal · Authors · 2023-08-29
>
> ### Comment
>
> Thank you for your valuable feedback! We follow all your suggestions and we believe we have solved all your concerns as follows! Given our new experimental results and explanations, we are happy to see you may reconsider the soundness of our work.
>
> ### Is it better to add explicit instructions for baselines?
>
> We have **already** used prompts with explicit instruction (i.e. “*the more interesting plots the better*”) to instruct the LLM to generate more plots for all our baselines.
>
> The Instruction in Table 3 of our submitted paper demonstrates an exemplary prompt to generate the initial story. The full prompt template is ***Prompt1***:
>
> >  [RETRIEVED EXAMPLE]. Learn from the plots and subjects in the given example, please write a [GENRE] that makes the readers feel [EMOTION]. It describes the following subjects: [SUBJECTS]. It should at least contain the following plots (the more interesting plots the better): [PLOTS].
>
> We have used the same prompt template to generate stories for CoT, ICL, and the initial stories for GROVE for a fair comparison. We will revise our paper to include these details in the experimental settings.
>
> We also conduct prompt engineering to further evaluate the generation ability of LLM. The results in Table 1 show that GROVE outperforms **Prompt Engineering** in almost all metrics.
>
> #### Table 1: Automatic evaluation of more suggested baselines.
>
> | Methods\Metrics    | Grammar       | Cherence| Likability    | Relevance     | Complexity    | Creativity    |
> | ------------------ | ------------- | ------------- | ------------- | ------------- | ------------- | ------------- |
> | Prompt Engineering | 4.97$_{0.18}$ | 4.32$_{0.58}$ | 3.76$_{0.47}$ | 4.37$_{0.84}$ | 3.30$_{0.59}$ | 3.31$_{0.63}$ |
> | GROVE              | 4.97$_{0.15}$ | 4.61$_{0.49}$ | 4.08$_{0.41}$ | 4.22$_{1.13}$ | 3.66$_{0.50}$ | 3.50$_{0.55}$ |
>
> **Prompt Engineering** instructs the LLM with different four phrases that explicitly say: “generate a complex and creative story”, “generate a detailed and complex story”, “Ensure that the story is creative and rich in plots”, and “generate a long and complex story”, respectively. We prompt the LLM with these instructions and take the average of their performance on the automatic evaluation.
>
> ### What is the prompt for the CoT baseline?
>
> We have briefly introduced our implementation of CoT in Line 480 of our submitted paper. We apologize for not listing the involved prompts. CoT first generates an initial story using ***Prompt1***. Then, it queries the LLM to revise the generated story step-by-step. The prompt template is ***Prompt2***, where $\mathcal{N}$ is 2 in our experiments:
>
> > After you write the story, point out the unclarities in the story in an organized list. Then provide further details to address the unclarities for $\mathcal{N}$ rounds. At last, integrate the details into the original story and start with the identifier "Integrated Story".
>
> We will list **all prompts** involved in all compared methods in our revised paper to avoid misunderstandings. Thank you.
>
> ### Can we prompt CoT to include the missing background information?
>
> Yes, actually, we **have implemented** the CoT baseline in our submitted paper to instruct the LLM to identify the missing information and iteratively revise the story to include missing backgrounds. We will articulate the implementation of the CoT baseline to avoid such misunderstanding.
>
> ### Is 16k model better?
>
> We experiment on the gpt-3.5-turbo-16k model and report the evaluation results in Table 2. We find that it has a higher possibility (more than 70%) to strictly follow the instructions in the CoT setting. However, it underperforms GROVE.
>
> #### Table 2: Automatic evaluation of gpt-3.5-turbo-16k model.
>
> | Methods\Metrics    | Grammar       | Cherence| Likability    | Relevance     | Complexity    | Creativity    |
> | --------------- | ------------- | ------------- | ------------- | ------------- | ------------- | ------------- |
> | CoT-16k         | 4.96$_{0.16}$ | 4.28$_{0.88}$ | 3.72$_{0.67}$ | 4.03$_{1.36}$ | 3.48$_{0.68}$ | 3.38$_{0.85}$ |
> | GROVE           | 4.97$_{0.15}$ | 4.61$_{0.49}$ | 4.08$_{0.41}$ | 4.22$_{1.13}$ | 3.66$_{0.50}$ | 3.50$_{0.55}$ |
>
> ---
>
> ### Explanation of  the name “asking why”, “evidence chain/forest”
>
> We call the prompting method “asking why” because it requires the LLM to justify **why** particular ambiguities make sense in the generated story.
>
> “Evidence chain” is inspired by the linked list in which one node connects to another because the latter evidence supports the former one in the evidence chain. “Evidence forest” is inspired by the data structure forest which contains multiple trees. In an evidence tree, the root node represents an ambiguous or unclear part of the generated story, while the non-root nodes represent additional information that provides clarity and background details to the nodes above them in the tree. We will define these concepts more clearly in the Introduction of our revised paper to make it more understandable.
>
> As the story itself is generated by an LLM, we presume it is acceptable to rely on the LLM to provide further backgrounds to make the story more sound. It is a good idea to exploit other data sources (e.g., a story database) to find valuable information to help refine the story. We would like to explore this idea in our future work!

---

### Official Review · Reviewer_xerB · 2023-08-05

**Soundness:** 4

**Excitement:**

3: Ambivalent: It has merits (e.g., it reports state-of-the-art results, the idea is nice), but there are key weaknesses (e.g., it describes incremental work), and it can significantly benefit from another round of revision. However, I won't object to accepting it if my co-reviewers champion it.

**Missing References:**

- The topic of this paper seems to be related to the task of story ending generation. The task was introduced in “﻿From Plots to Endings: A Reinforced Pointer Generator for Story Ending Generation” [Zhao et al., NLPCC 2018]”, and there are representative works such as “Towards Controllable Story Generation” [Peng et al.,  the First Workshop on Storytelling, 2018], “Story Ending Generation with Incremental Encoding and Commonsense Knowledge” [Guan et al., AAAI 2019].

- The year information in (Aho and Ullman, 1972) seems strange. The founding year of IMDB is 1990, so the year of the dataset must be later.

**Paper Topic And Main Contributions:**

To generate complex and creative plots for story generation, this paper proposed a retrieval-auGmented stoRy generation framework with fOrest of eVidEnce (GROVE). GROVE consists of three parts: Retrieval Repository, Evidence Forest Construction via Asking Why, and Evidence Chains-supported Story Rewriting. GROVE achieved SOTA performance.

**Questions For The Authors:**

Question A: Have you conducted an experiment of interactive generation, where human writers are involved not in scoring but in collaborative writing?

Question B: As discussed in the Ethics Statement, there is the possibility of violating copyright laws in such a system. Is there any idea of avoiding such a problem?

Question C: Can the proposed method be applied to a situation where writers intentionally create ambiguity? For example, some stories require readers to read between the lines.

Question D: What version of ChatGPT do you use? It seems the result depends on it.

Question E: It is stated that GROVE can be based on various LLMs, but Ii seems you use the same LLM for all components. Have you tried using different LLMs in one GROVE? In other words, can each part of the three explained in 3.1 use different LLMs?

**Reasons To Accept:**

Tackling the challenging topic of having both controllability and creativity, the proposed method achieved a good result.

**Reasons To Reject:**

Some information is not fully explained. The term “a forest of evidence” (or “evidence forest”) should be clearly defined (or cited) because this term seems to be one of the key points of this paper.
Fig. 1 is explained in Section 3.1, but there is no explanation about where is the corresponding part in 3.1 and steps 1 to 3.

**Reproducibility:**

4: Could mostly reproduce the results, but there may be some variation because of sample variance or minor variations in their interpretation of the protocol or method.

**Reviewer Confidence:**

3: Pretty sure, but there's a chance I missed something. Although I have a good feel for this area in general, I did not carefully check the paper's details, e.g., the math, experimental design, or novelty.

**Typos Grammar Style And Presentation Improvements:**

- Line 084: missing white space before “To.”

- The format of quotation marks is not uniform. Most of them (lines 106, 278, 282, 283, 285, 286, 330, 343, 348, 352, 367, 373, 405, 412) are curved, but some (lines 196, 197, 590, 592) are straight.

---

> ### Author Rebuttal · Authors · 2023-08-29
>
> ### Comment
>
> Thank you for your thoughtful review and interesting questions! Here we address your questions about our work.
>
> ### The explanation about the evidence forest
>
>  We will provide a clear and concise definition of the term "evidence forest" in the revised version.
>
> An evidence forest refers to a collection or set of evidence trees that are generated to supplement a story in GROVE. Each evidence tree consists of nodes representing pieces of evidence and edges connecting them. The root node of the tree represents an ambiguous or unclear part in the generated story, while the non-root nodes represent additional information that provides clarity and background details to the nodes above them in the tree.
>
> We will articulate its definition at the beginning of the paper and cite the corresponding part in Fig 1 to ensure a thorough understanding by the readers.
>
> ### The explanation of Fig 1
>
>  In our revised paper, we will explain the steps 1 to 3 and cite the corresponding parts of Fig 1 in Section 3.1.
>
> Step 1 is to query the retrieval repository with target conditions for examples. Steps 2 and 3 is to prompt the LLM to generate an initial story with target conditions and retrieved examples.
>
> ### Q A: How about interactive generation?
>
> Thank you for your suggestion! We are unable to conduct interactive generation experiments on GROVE due to the time limit. We are happy to explore collaborative story writing in our future work!
>
> It may be a promising idea to let LLMs produce an initial story and summarize its plots, then allow human readers to question the rationale behind certain plot developments. Subsequently, we could instruct the LLM to modify the story by incorporating the requisite details that support the initial storyline.
>
> ### Q B: Any ideas regarding avoiding copyright violation?
>
> Copyright violation is a common issue for language model-based text generation systems$^{[1,2]}$. In our work, we have conducted plagiarism detection on GROVE in our submitted paper (Appendix A). using a commercial AI-based text analysis tool (Copyleaks). Our experiment indicated that there is no evidence of plagiarism as we **detected 0% plagiarism in four categories** (Identical, Minor Changes, Paraphrased, and Omitted Words).
>
> To further enhance copyright protection, we suggest using licensed or permitted story datasets to inspire generating new stories to alleviate risks.
>
> ### Q C: Can we create ambiguity intentionally with GROVE?
>
> Yes, users can prompt the LLM to generate stories that contain implications or create suspense and instruct the model to retain important suspense during evidence chains-supported rewriting while filling in the details that decrease the credibility of the story.
>
> ### Q D: What is the version of ChatGPT in our experiment?
>
> We conducted our experiments on the default **gpt-3.5-turbo** model (see Appendix C).
>
> ### Q E: Can we use different LLMs in one GROVE
>
> Thank you for your suggestion! We have verified that all steps in GROVE work on the gpt-3.5-turbo and Alpaca-plus-7B model (in Section 4.4), indicating the strong generalization ability of GROVE on varying sizes of LLMs. Therefore, we thought that GROVE would probably still work with different components based on various LLMs (e.g., use Alpaca to generate the initial story and use ChatGPT to make revisions).
>
> It is a good idea to use different LLMs in different steps. Due to the time limit, we will explore this idea and conduct experiments using different LLMs in one GROVE in the revised paper.
>
> ### Missing references and grammatical errors
>
> We will cite the missing references and resolve all grammatical errors in the revised version. Thank you!
>
> ## References
>
> [1] Jessica L Gillotte. 2019. Copyright infringement in ai-generated artworks. UC Davis L. Rev.
>
> [2] John Kirchenbauer, Jonas Geiping, Yuxin Wen, Jonathan Katz, Ian Miers, and Tom Goldstein. 2023. A watermark for large language models. arXiv preprint arXiv:2301.10226.

---

### Meta-Review · Area_Chair_k44e · 2023-09-18

**Recommendation:** 3

**Metareview:**

This paper proposes a retrieval-augmented strategy to enhance story generation. In short, based on some metadata (e.g., plot, genre, mood), similar plots from the IMDb dataset are retrieved, and then relevant info from these plots is then used to revise an initially-generated story. All reviewers praise the idea's creativity and the evaluation conducted by experts. However, there are concerns about the method's novelty (there is perhaps too much complexity in the prompting strategy) and performance (improvements directly attributed from the paper's main ideas are small). The authors provided many new experiments and details in their rebuttal / discussion which assuaged other complaints about generalizability, which they are to be commended for, and I'd encourage the authors to include human evaluations in the next version of the paper for the new configurations they tried in their rebuttal. Overall, definitely a good inclusion to Findings!

---

### Decision · Program_Chairs · 2023-10-07

**Decision:**

Accept-Findings

**Comment:**

This paper proposes a retrieval-augmented strategy to enhance story generation. In short, based on some metadata (e.g., plot, genre, mood), similar plots from the IMDb dataset are retrieved, and then relevant info from these plots is then used to revise an initially-generated story. All reviewers praise the idea's creativity and the evaluation conducted by experts. However, there are concerns about the method's novelty (there is perhaps too much complexity in the prompting strategy) and performance (improvements directly attributed from the paper's main ideas are small). The authors provided many new experiments and details in their rebuttal / discussion which assuaged other complaints about generalizability, which they are to be commended for, and I'd encourage the authors to include human evaluations in the next version of the paper for the new configurations they tried in their rebuttal. Overall, definitely a good inclusion to Findings!